# Ventricular divergence correlates with epicardial wavebreaks and predicts ventricular arrhythmia in isolated rabbit hearts during therapeutic hypothermia

Yu-Cheng Hsieh [1,2,3]*, Wan-Hsin Hsieh[4], Cheng-Hung Li[1,2,3], Ying-Chieh Liao[1,2,3], Jiunn-Cherng Lin[1,2,3], Chi-Jen Weng[1,2,3], Men-Tzung Lo[5], Ta-Chuan Tuan[2,6], Shien-Fong Lin [7,8], Hung-I Yeh[9], Jin-Long Huang[1,2], Ketil Haugan[10], Bjarne D. Larsen[11], Yenn-Jiang Lin[2,6], Wei-Wen Lin[1,12], Tsu-Juey Wu[1,2], Shih-Ann Chen[2,6]

1 Cardiovascular Center, Taichung Veterans General Hospital and Chiayi Branch, Taichung and Chiayi, Taiwan, 2 Department of Internal Medicine, Faculty of Medicine, Institute of Clinical Medicine, Cardiovascular Research Center, National Yang-Ming University School of Medicine, Taipei, Taiwan, 3 Department of Data Science and Big Data Analytics and Department of Financial Engineering, Providence University, Taichung, Taiwan, 4 Biomedical Technology and Device Research Laboratories, Industrial Technology Research Institute, Hsinchu, Taiwan, 5 Research Center for Adaptive Data Analysis, National Central University, Jhongli City, Taiwan, 6 Division of Cardiology, Department of Medicine, Taipei Veterans General Hospital, Taipei, Taiwan, 7 Krannert Institute of Cardiology and the Division of Cardiology, Department of Medicine, Indiana University School of Medicine, Indianapolis, IN, United States of America, 8 Institute of Biomedical Engineering, National Chiao-Tung University, Hsinchu, Taiwan, 9 Departments of Internal Medicine and Medical Research, Mackay Memorial Hospital, Mackay Medical College, New Taipei City, Taiwan, 10 Department of Cardiology, Zealand University Hospital, Roskilde, Denmark, 11 Zealand Pharma A/S, Glostrup, Denmark, 12 Department of Life Science, Tunghai University, Taichung, Taiwan

* ychsieh@vghtc.gov.tw

**Data Availability Statement:** All relevant data are within the manuscript and its Supporting Information files.

## Abstract

### Introduction

High beat-to-beat morphological variation (divergence) on the ventricular electrogram during programmed ventricular stimulation (PVS) is associated with increased risk of ventricular fibrillation (VF), with unclear mechanisms. We hypothesized that ventricular divergence is associated with epicardial wavebreaks during PVS, and that it predicts VF occurrence.

### Method and results

Langendorff-perfused rabbit hearts (n = 10) underwent 30-min therapeutic hypothermia (TH, 30˚C), followed by a 20-min treatment with rotigaptide (300 nM), a gap junction modifier. VF inducibility was tested using burst ventricular pacing at the shortest pacing cycle length achieving 1:1 ventricular capture. Pseudo-ECG (p-ECG) and epicardial activation maps were simultaneously recorded for divergence and wavebreaks analysis, respectively. A total of 112 optical and p-ECG recordings (62 at TH, 50 at TH treated with rotigaptide) were analyzed. Adding rotigaptide reduced ventricular divergence, from 0.13±0.10 at TH to 0.09±0.07 (p = 0.018). Similarly, rotigaptide reduced the number of epicardial wavebreaks, from 0.59±0.73 at TH to 0.30±0.49 (p = 0.036). VF inducibility decreased, from 48±31% at

**Funding:** This study was supported by grants from the National Science Council (104-2314-B-367-001 and 105-2314-B-075A-016 -MY3), Taipei, Taiwan; and Taichung Veterans General Hospital (TCVGH-1033105C, TCVGH-1043109C, TCVGH-1053108C, VGHUST104-G5-2-2, VGHUST107-G5-1-3, TCVGH-VHCY1068606, and TCVGH-VHCY1078603), Taichung, Taiwan. Zealand Pharma A/S provided support in the form of salaries for author [BDL] and rotigaptide, but did not have any additional role in the study design, data collection and analysis, decision to publish, or preparation of the manuscript.

**Competing interests:** Zealand Pharma A/S provided support in the form of salaries for author [BDL] and rotigaptide. This does not alter our adherence to PLOS ONE policies on sharing data and materials.

TH to 22±32% after rotigaptide infusion (p = 0.032). Linear regression models showed that ventricular divergence correlated with epicardial wavebreaks during TH (p<0.001).

## Conclusion

Ventricular divergence correlated with, and might be predictive of epicardial wavebreaks during PVS at TH. Rotigaptide decreased both the ventricular divergence and epicardial wavebreaks, and reduced the probability of pacing-induced VF during TH.

## Introduction

Sudden cardiac arrest (SCA) from ventricular arrhythmia (VA) constitutes a major public health problem and accounts for approximately 50% of all cardiovascular deaths [1]. Despite the progress been made in risk prediction of SCA, the greatest challenge is to identify reliable risk predictors in patients with and without structural heart disease [2]. Beat-to-beat morphological variation (divergence) in the duration and amplitude of ventricular action potentials has been recognized as a promising index of arrhythmic susceptibility and dynamic electrical instability that predisposes to ventricular tachycardia (VT) and fibrillation (VF) [3]. Such subtle morphological variation might manifest as T wave and microvolt T wave alternans in ECG, which has been correlated with the risk for VA and SCA [4].

In patients with an implantable cardioverter-defibrillator (ICD), beat-to-beat morphological variation might even be recorded by ICD leads before spontaneous VT and VF, suggesting that these morphological variations might serve as a warn sign before VF and subsequent ICD shocks [5]. In patients who were unable to achieve an adequate heart rate for beat-to-beat morphological variation evaluation, programmed ventricular stimulation (PVS) using burst ventricular pacing by electrodes or the atrial/ventricular leads of an implantable device has been found to be a reliable alternative to predict VA [6]. By analyzing the electrograms from the anti-tachycardia pacing episodes in the ICD, we also found that high ventricular divergence is predictive of a failed therapy with a sensitivity and specificity of 81.9% and 65.9%, respectively [7]. Although morphological variation has been liked to temporal dispersion of repolarization in myocardial substrate, the mechanism by which the increased divergence leads to VA remains unclear [8].

Epicardial wavebreaks resulting from multiple wave collisions may predispose the myocardial substrate to fibrillate, serving as a potential source of VF [9, 10]. We reported earlier that therapeutic hypothermia (TH) at 30°C could create an arrhythmia substrate and increase epicardial wavebreaks for VF maintenance in isolated rabbit hearts [11–13]. The infusion of the gap junction modifier rotigaptide during TH could enhance ventricular conduction, decrease epicardial wavebreaks, and reduce the incidence of VF [11]. Smith et al also found that TH at 29°C increased the beat-to-beat ECG variation during burst ventricular pacing, which was associated with more inducible VT/VF in a canine experiment [14]. With this pro-arrhythmic model, we were able to evaluate the myocardial substrate properties by delivering burst ventricular pacing, while simultaneously measuring the divergence and epicardial wavebreaks before and after rotigaptide infusion. We hypothesized that high ventricular divergence at ECG during PVS is associated with more epicardial wavebreaks, leading to the occurrence of pacing-induced VF (PIVF). Enhancing cell-to-cell coupling using rotigaptide during TH may simultaneously reduce ventricular divergence and epicardial wavebreaks, thus reducing the probability of PIVF. Understanding this mechanism might improve the patient safety

during TH by delivering essential anti-arrhythmic therapy when increased beat-to-beat morphological was detected before the recurrence VF.

## Methods

The study protocol was approved by the Institutional Animal Care and Use Committee of Taichung Veterans General Hospital (La-1051352).

### Langendorff preparation and pseudo-ECG recordings

New Zealand white rabbits (3.0±0.3 kg, n = 19) were used. Before anesthetization, ketamine (10 mg/kg) was injected intramuscularly to calm the animals. After 10–15 min, the rabbits were intravenously injected with heparin (1,000 units) and anesthetized with sodium pentobarbital (35 mg/kg) via the marginal ear vein. After a median sternotomy, the hearts were rapidly excised. The ascending aorta was cannulated and perfused with 37°C oxygenated Tyrode's solution composed of (in mM): 125 NaCl, 4.5 KCl, 0.5 MgCl$_2$, 24 NaHCO$_3$, 1.8 NaH$_2$PO$_4$, 1.8 CaCl$_2$, 5.5 glucose, and albumin (40 mg/L) in deionized water [11–13]. Coronary perfusion pressure and flow rate were maintained at 60–65 mmHg and 35–45 ml/min, respectively [11–13]. The hearts were also superfused in a thermostatized tissue bath. Pseudo-ECG (p-ECG) was obtained through use of a pair of electrodes attached to both ventricles to determine the ventricular rhythm [11]. A bipolar hook electrode was anchored to the right ventricular outflow tract (RVOT) for ventricular pacing.

### Optical mapping

Epicardial activations in the anterior and posterior surfaces of the hearts were simultaneously mapped by a two-camera optical mapping system [11–13]. The hearts were stained with di-4-ANEPPS, and excited with 4 light-emitting diode modules (wavelength = 519±20 nm). Induced fluorescence was collected by two image-intensified charge-coupled cameras (model CA D1-0128T) [11–13]. Optical signals were gathered at 3.85-ms sampling intervals for 1,000 frames, acquired from 128x128 sites simultaneously over a 30x30 mm$^2$ area in each aspect of the heart. For each recording, optical data was acquired continuously for 3.85 sec (1,000 phase maps). In previous studies, we have demonstrated that a similar camera system with this frame rate could successfully capture wavebreaks and phase singularities (PSs) [15, 16]. Furthermore, a long exposure time (3.85 sec) is essential to capture the whole process from the beginning of burst pacing to the initiation of VF in optical mapping system. Cytochalasin-D (5 μM) was used as an excitation-contraction un-coupler to minimize motion artifacts. In a typical time-embedded phase portrait, the upstroke of the action potential corresponds to a phase ranging from $-3/4\pi$ to $-1/4\pi$, displayed in light-blue pixels on the representing color scale [11–13].

### Induction of hypothermia (30°C)

Two thermostatic systems were used to control the temperature of the hearts [11, 13]. By switching between the two systems, the tissue bath temperature could be inter-changed between warm (37°C) and cool (30°C) settings. To induce hypothermia, we shifted the thermostatic system to the cool setting, and at the same time with the superfusate quickly replaced with a cool Tyrode's solution. During the cooling procedure, the temperature in the upper, middle, and lower thirds of the tissue bath was checked every 1–2 min until 30°C was achieved at all levels. After reaching the target temperature (30°C), the tissue bath was kept at this temperature for an additional 5 min to ensure thermal homogeneity, before starting the study protocol.

## Study protocol

**Protocol I: Rotigaptide on ventricular divergence, epicardial wavebreaks, and susceptibility to VF during TH (30˚C) (n = 10).** $S_1$ pacing (2× diastolic threshold) protocol was used to determine the ventricular conduction velocity (CV), action potential duration (APD), and susceptibility to VF [11]. CV and APD were obtained with pacing cycle lengths (PCL) 400 to 120 ms at each stage. At baseline (37˚C), we progressively shortened the $S_1$ PCL (frequency) to identify the shortest PCL which achieved a 1:1 ratio ventricular capture. This PCL was selected as the initial PCL for burst pacing to induce VF. Afterwards, the heart was cooled and maintained stably at 30˚C for 30 min. Five to ten burst pacing trains (30 sec in duration) using the shortest PCL for ventricular capture were delivered to test the inducibility of VF. If VF could be induced and persisted for >1 min, a defibrillation shock would be delivered through the defibrillation coil, and this episode was considered to be a PIVF episode [17]. Afterwards, rotigaptide (300 nM) was infused into the heart for 20 min, and VF induction protocol was repeated. We had chosen the concentration of rotigaptide to be 300 nM based upon the maximal effective concentration reported in the literature [18]. During the burst pacing period in each VF inducibility test, both optical (for wavebreak evaluation) and p-ECG (for divergence) data were simultaneously recorded.

**Protocol II: Vehicle on electrophysiological properties and susceptibility to VF during TH (30˚C) (n = 4).** This protocol is the same as protocol I, except for using vehicle (saline) instead of rotigaptide.

**Protocol III: Different pacing sites on ventricular divergence, epicardial wavebreaks and susceptibility to VF during TH (30˚C) (n = 3).** Another bipolar hook electrode was placed at the lateral aspect of left ventricle (LV), in addition to the original one at RVOT. This protocol is the same as protocol I, except for that in each stage, ventricular pacing was performed at RVOT firstly followed by pacing from LV.

**Protocol IV: Extensive endocardial ablation on ventricular divergence, epicardial wavebreaks and susceptibility to VF during TH (30˚C) (n = 2).** This protocol is the same as protocol I, except for using Lugol's solution (5 g $I_2$ and 10 g KI dissolved in 100 ml deionized $H_2O$) to paint the endocardial surface of both ventricles with 2 cotton swabs for 10–15 sec to perform extensive endocardial ablation before starting the experimental protocol.

## Data analysis

**Construction of CV and APD restitution curves using $S_1$ pacing method.** CV and APD and restitution curves were constructed as previously reported [11, 12]. Briefly, epicardial activation perpendicular to the propagating wavefront was selected to measure the CV. The epicardial CV was evaluated by dividing the distance between 2 epicardial points with the conduction time (CT) using depolarization isochronal maps (Fig 1A). The CT between 2 epicardial points was measured by 50% crossover of the action potential amplitude in activation maps. We evaluated the CV at the centers of the anterior (A) and posterior (P) aspects of both ventricles (RV and LV). The mean of the CVs from these 4 areas became the CV of the heart. For APD evaluation, pixels at the center of the anterior and posterior surface of both ventricles were selected to determine the $APD_{80}$ (APD at 80% repolarization) and APD restitution curves. The maximal slope of APD restitutions for each heart is the mean of the data from these 4 sampling sites.

Phase mapping was performed to evaluate the wavefront characteristics, along with the location and evolution of the PSs [12, 17]. A PS is defined as a site with an ambiguous phase surrounded by pixels exhibiting a continuous phase progression from −π to +π. Because of the close spatiotemporal correlation between PSs and wavebreak, PS has been a robust alternate

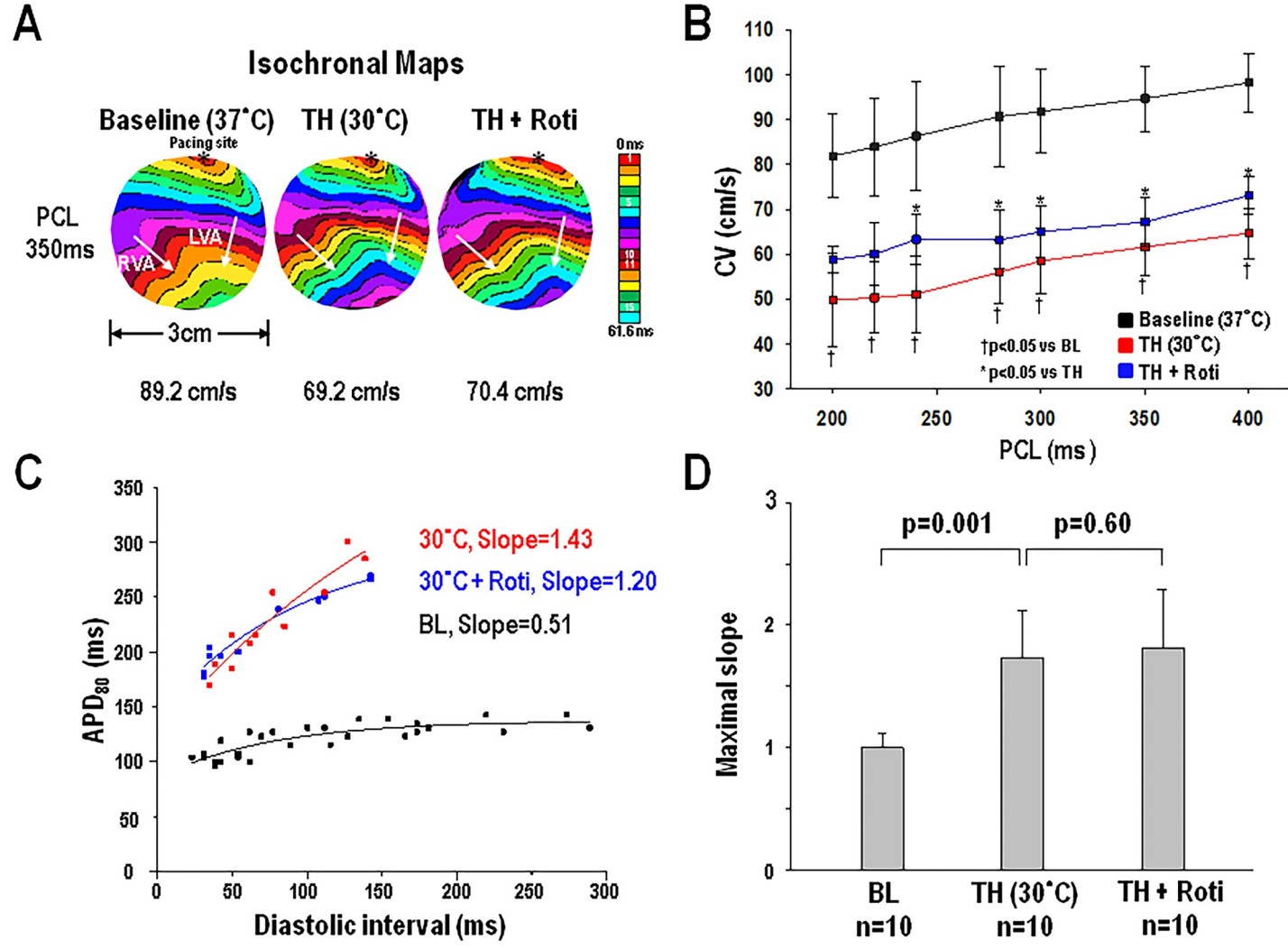

**Fig 1.** A, representative isochronal maps (from heart #10) at pacing cycle length (PCL) 350 ms during baseline (BL, 37˚C), therapeutic hypothermia (TH, 30˚C), and after rotigaptide (Roti) infusion. These maps are constructed by consecutive frames shown in different colors (1 frame = 3.85 ms), and used for evaluation of ventricular conduction velocity (CV). The average CVs of each stage are shown below. *Pacing site. B, CVs during these 3 stages at different PCLs. C, representative action potential duration (APD) restitution curves from a site at left ventricle (from heart #9) during these 3 stages. D, maximal slope of APD restitutions at there 3 stages.

representations of wavebreaks, serving as the source of VF [10]. By screening through phase maps frame-by-frame, epicardial PSs were identified. To quantify wavebreaks during VF, the number of PSs in each phase map was counted manually every 10 frames over the course of 1,000 frames in each optical recording [12]. Fast Fourier Transforms of p-ECGs (4 sec in duration) were used to determine the dominant frequency (DF) of each VF studied.

APD alternans was defined as the difference in $APD_{80}$ on two consecutive beats of $\geq 3.85$ ms during $S_1$ pacing [11]. Spatially concordant alternans (SCA) was defined as APD alternation in phase spatially, whereas spatially discordant alternans (SDA) was defined as an alternation out of phase spatially [19]. Therefore, the SDA threshold was defined as the longest $S_1$ PCL at which SDA was detected.

We have previously used the beat-to-beat morphologic variation (divergence) to predict the response of anti-tachycardia pacing on VT in patients with structural heart disease [7]. Note that the term of divergence is also known for the measure of the "distance" of one probability

distribution to the other in statistics, which is also known as "relative entropy" [20]. We employed the concept to analyze the degree of morphologic variation of each activation on p-ECG in this study. The maximal activation (local peak) of each p-ECG was identified firstly. Each p-ECG was then normalized with respect to the maximum values. The length of each p-ECG was defined by 80% of the average duration between the two consecutive activations. Each p-ECG was then extracted and aligned by the point of maximal activation so as to prevent the morphologic variation from the interference caused by the varying interval of two con-secutive activations and the varying signal strength. Finally those aligned waveforms were applied to the proposed divergence equation. The $i$th normalized p-ECG can be denoted by $X_i = \left\{ s_{k-\frac{L+1}{2}} \ldots s_k \ldots s_{k+\frac{L-1}{2}} \right\}$, where $L$ is the data length of the $i$th p-ECG, and $s_k$ is the data point with the maximum voltage. A template p-ECG $\bar{X}$ was constructed by averaging all of the normalized p-ECG, i.e. $\bar{X} = \frac{1}{N} \sum_{i=1}^{i=N} X_i$ for an episode of $N$ p-ECG. The morphologic variation between the $i$th p-ECG and template was derived by calculating the standard deviation (std) of the difference between $i$th p-ECG and template, i.e. $d(X_i, \bar{X}) = std(X_i - \bar{X})$, for which is used to reflect the averaged 2-norm distance between the $i$th p-ECG and template. The overall mor-phologic variation was finally obtained by averaging all of the morphologic variations regard-ing the $i$th p-ECG, which is the "divergence" of the p-ECG, i.e., $Divergence = \frac{1}{N} \sum_{i=1}^{N} d(X_i, \bar{X})$ [7]. Note that the divergence will be zero for an episode with identical ventricular p-ECG.

## Statistical analysis

Data were presented as mean± SD. We used the non-parametric Wilcoxon signed-rank test for paired comparisons (within the groups), and the Mann-Whitney U test for independent comparisons (between the groups). Analysis of variance (ANOVA) was used to compare the means of three or more groups, with p-values adjusted by the Bonferroni method. Fisher's exact test was used to compare the categorical data between groups. Probability values of p $\leq$ 0.05 were considered statistically significant.

## Results

### Rotigaptide on CV and APD restitutions during TH (30°C)

The cycle lengths of baseline spontaneous beats during TH were 1153±262 ms at the beginning of the protocol, and 1182±385 ms before rotigaptide treatment. The effects of rotigaptide on CV and APD restitutions during TH were shown in Fig 1. Compared to baseline, CVs were decreased during TH (Fig 1A and 1B). Rotigaptide increased the CVs during TH (i.e., at PCL 350 ms, TH: 61.6±6.3 cm/s, TH+Roti: 67.2±5.5 cm/s, p = 0.012) (Fig 1B). Compared to base-line, APDs were prolonged during TH. However, rotigaptide did not change the APDs during TH (i.e., at PCL 350 ms, TH: 238±12 ms, TH+Roti: 243±15 ms, p = 0.28). The maximal slope of APD restitution was increased from 1.00±0.12 at baseline to 1.73±0.39 during TH (p = 0.001) (Fig 1C and 1D). However, the maximal slope of APD restitution was not changed by rotigaptide during TH (p = 0.60). In control group, rotigaptide did not change APDs (i.e., at PCL 350 ms, TH: 232±14 ms, TH+vehicle: 238±9 ms, p = 0.22), CV (i.e., at PCL 350 ms, TH: 63.7±16.4 cm/s, TH+vehicle: 63.8±19.9 cm/s, p = 0.98) and maximal slope of APD restitutions (TH: 2.01±0.57, TH+vehicle: 2.26±1.23, p = 0.70).

### Effects of rotigaptide on VF inducibility during TH

Of the 10 hearts in the rotigaptide group, a total of 112 optical/p-ECG recordings (62 at TH, 50 at TH plus rotigaptide) during burst ventricular pacing were analyzed. The experiment

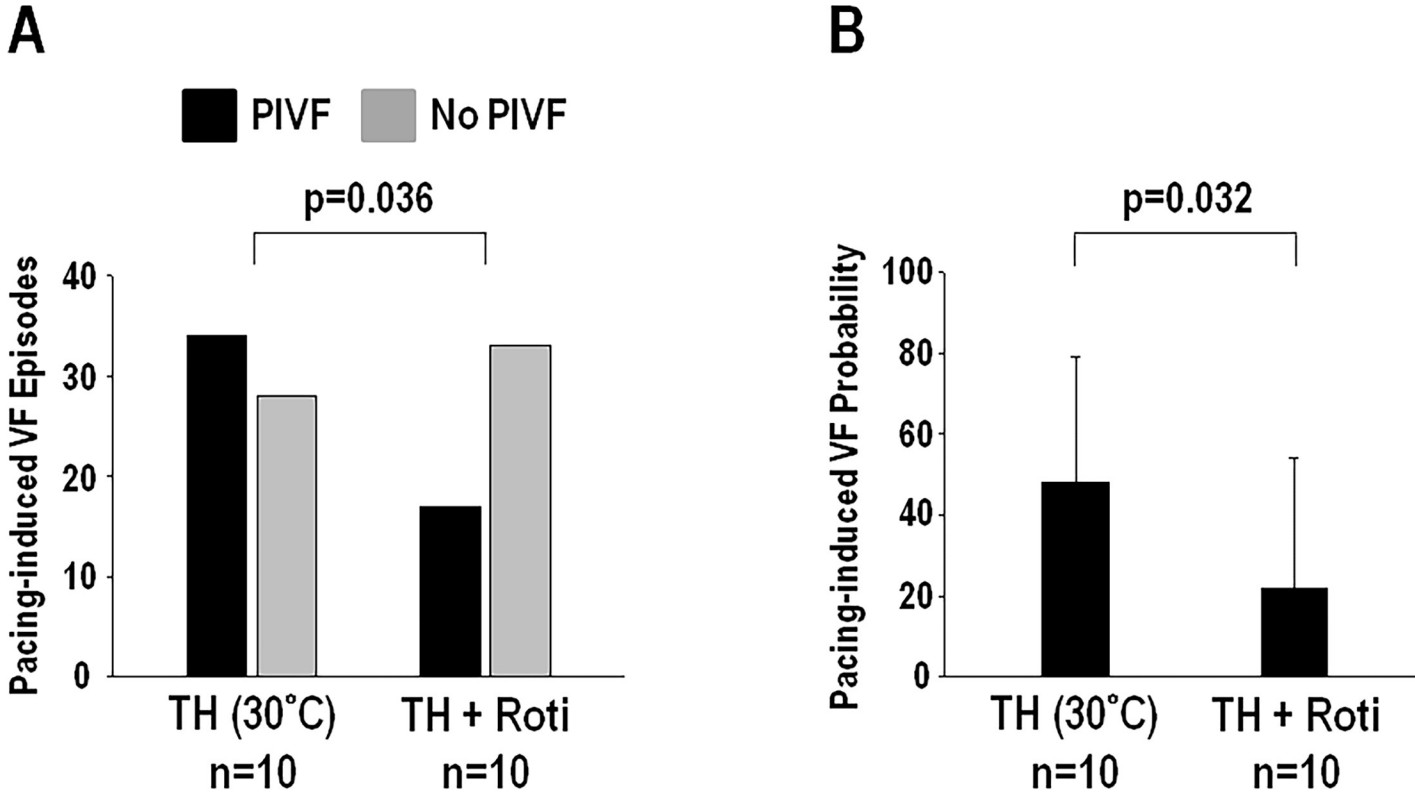

**Fig 2.** A, Rotigaptide (Roti) on pacing-induced ventricular fibrillation (VF) episodes during therapeutic hypothermia (TH, 30˚C). B, Rotigaptide on the probability of pacing-induced VF during TH.

duration of each heart was 126±15 min (S1 Fig). S1 Table showed the PCLs and results of each pacing attempt in the rotigaptide group. Two hearts (#1 and #6) did not develop any PIVF episode throughout the pacing protocol. The PCLs used for VF induction were similar both before (206±40 ms, n = 62) and after (204±46 ms, n = 50) rotigaptide during TH (p = ns). During TH, rotigaptide decreased the percentage of PIVF episodes from 55% (34 out of 62 episodes) to 34% (17 out of 50 episodes) (p = 0.036) (Fig 2A). The PCL used in PIVF episodes (196±37 ms, n = 51) is shorter than that in non-PIVF episodes (212±45 ms, n = 61, p = 0.044). In the PIVF episodes, the DF of the VFs were indifferent before (6.7±1.5 Hz, n = 34) and after (6.2±1.1 Hz, n = 17) rotigaptide infusion (p = 0.20). The VF inducibility also decreased from 48±31% at TH to 22±32% after rotigaptide infusion (p = 0.032) (Fig 2B). In control group, the VF inducibility was not changed with vehicle infusion (TH: 25±25%, TH+vehicle: 58±50%, p = 0.16).

### Rotigaptide on ventricular divergence and epicardial wavebreaks during TH

Fig 3 shows an example of ventricular divergence both before (Fig 3A) and after rotigaptide infusion (Fig 3B) during TH. The ventricular divergence was higher in the PIVF episodes (0.16 ±0.11, n = 51) than that of the non-PIVF episodes (0.09±0.07, n = 61) (p<0.001) (Fig 4A) during TH. Rotigaptide decreased the ventricular divergence from 0.13±0.10 (n = 62) at TH to 0.09±0.07 (n = 50) after being added rotigaptide (p = 0.018) (Fig 4B). Similar to the changes in divergence, the PIVF (0.85±0.71) episodes had a higher number of epicardial wavebreaks than those of the non-PIVF (0.11±0.26) episodes during TH (p<0.001) (Fig 4C). The number of

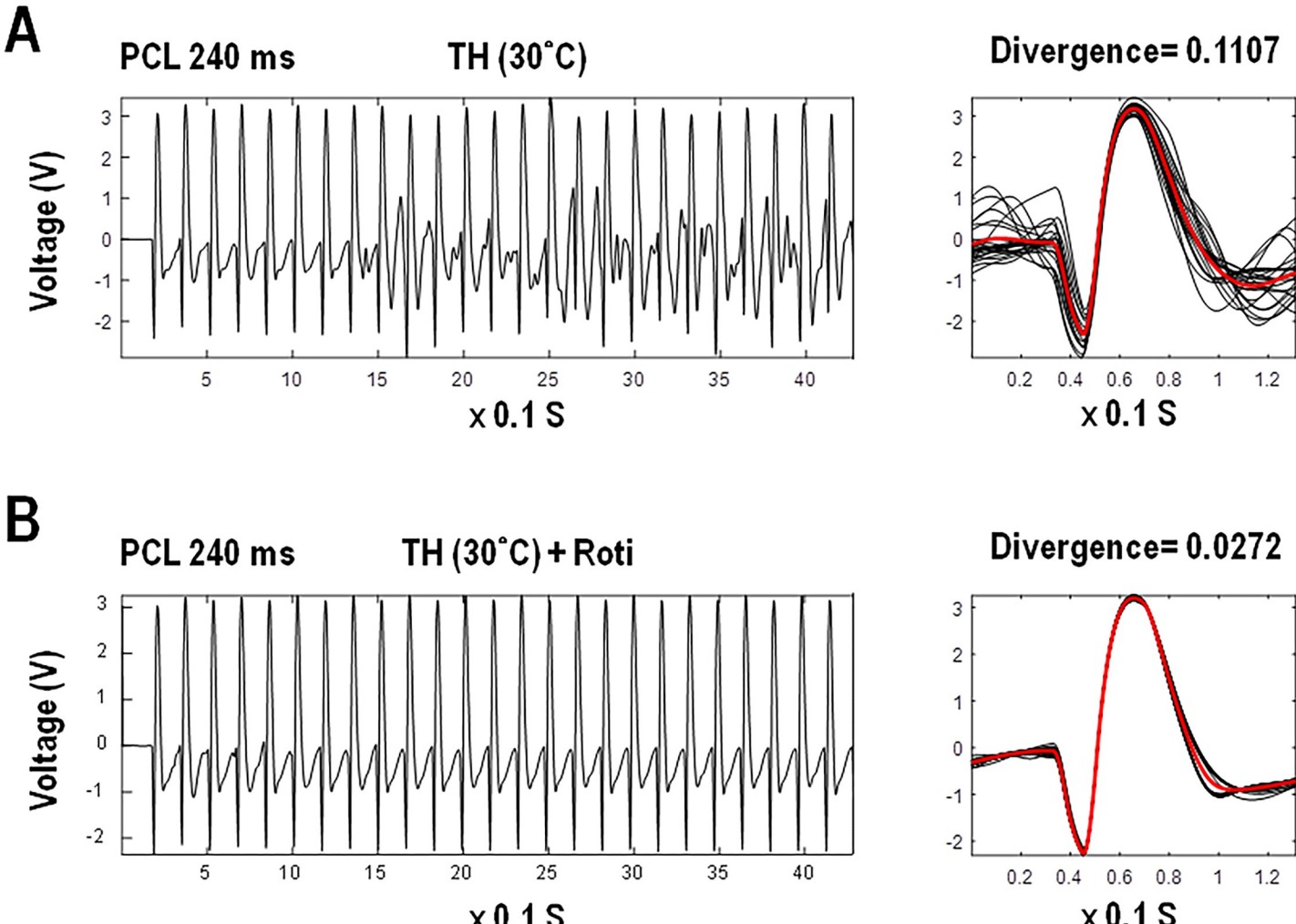

**Fig 3. Two examples of divergence analysis at burst ventricular pacing cycle length (PCL) of 240 ms during therapeutic hypothermia (TH, 30˚C) in heart #10.** A, without rotigaptide, this episode shows significant morphological variation during pacing with a high divergence of 0.1107. VF occurred immediately after stopping pacing. B, with rotigaptide, this episode shows minimal morphological change during pacing with a low divergence of 0.0272. Normal rhythm was observed immediately after stopping pacing.

epicardial wavebreaks also dropped from 0.59±0.73 at TH to 0.30±0.49 after rotigaptide treatment (p = 0.036) (Fig 4D). The life spans of each wavebreak in PIVF episodes were similar before (102±48 ms) and after (91±32 ms) rotigaptide (p = 0.50).

Fig 5 shows the dynamic changes in beat-to-beat ventricular divergence and epicardial wavebreaks in PIVF and non-PIVF episodes, as well as the changes with and without rotigaptide. In these episodes, divergence abruptly increased during burst pacing, and then reached a steady state level after the 6th beat (Fig 5A). Fig 5B shows that the PIVF episodes have a higher divergence than those of the non-PIVF episodes with (PIVF: 0.14±0.10, non-PIVF: 0.07±0.03, p = 0.02) or without (PIVF: 0.16±0.11, non-PIVF: 0.10±0.05, p = 0.004) rotigaptide treatment. In the non-PIVF episodes, the divergence was lower when rotigaptide was introduced (0.07 ±0.03), than it was without (0.10±0.05) during TH (p = 0.04). In the PIVF episodes, the divergence was unaffected by the rotigaptide infusion (p = 0.41). Similar to the changes in divergence, the epicardial wavebreak number progressively increased during pacing in the PIVF episodes, while the wavebreak numbers remained few in the non-PIVF episodes (Fig 5C). The

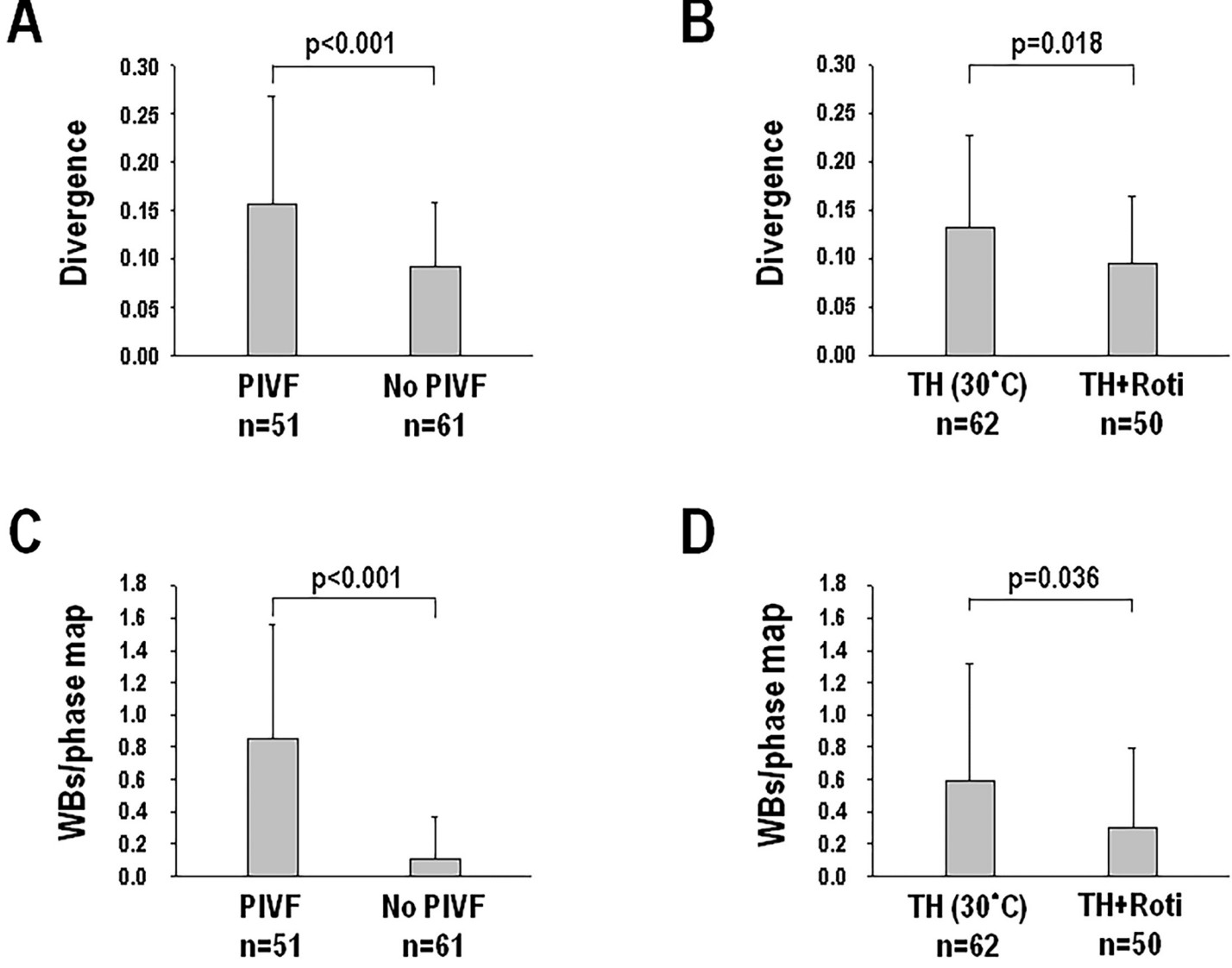

**Fig 4.** Ventricular divergence (A, B) and epicardial wavebreaks (C, D) in the pacing-induced ventricular fibrillation (PIVF) and non-PIVF (no VF) episodes with and without rotigaptide (Roti) infusion during therapeutic hypothermia (TH, 30˚C).

PIVF episodes also had more epicardial wavebreaks than those of the non-PIVF episodes, both with (PIVF: 0.86±0.53, non-PIVF: 0.05±0.10, p<0.001) and without (PIVF: 0.85±0.80, non-PIVF: 0.22±0.40, p = 0.005) rotigaptide (Fig 5D). In non-PIVF episodes, the wavebreak number was lower with rotigaptide (0.05±0.10) than it was without rotigaptide (0.22±0.40) during TH (p = 0.03). In the PIVF episodes, the wavebreak number was similar with or without rotigaptide (p = 0.96).

Fig 6 provides an example of simultaneous epicardial activation and ventricular divergence during ventricular pacing in a rotigaptide-treated non-PIVF episode. The ventricular divergence was low, and epicardial wavebreaks were also few during pacing. Conversely, Fig 7 is an example of a PIVF episode in the absence of rotigaptide during TH. In this episode, the ventricular divergence and epicardial wavebreak numbers were both high during pacing. Unidirectional block and wavebreak occurred soon during pacing with sustained reentry degenerated into complex wavefront collisions and fibrillatory conduction (S1 Movie).

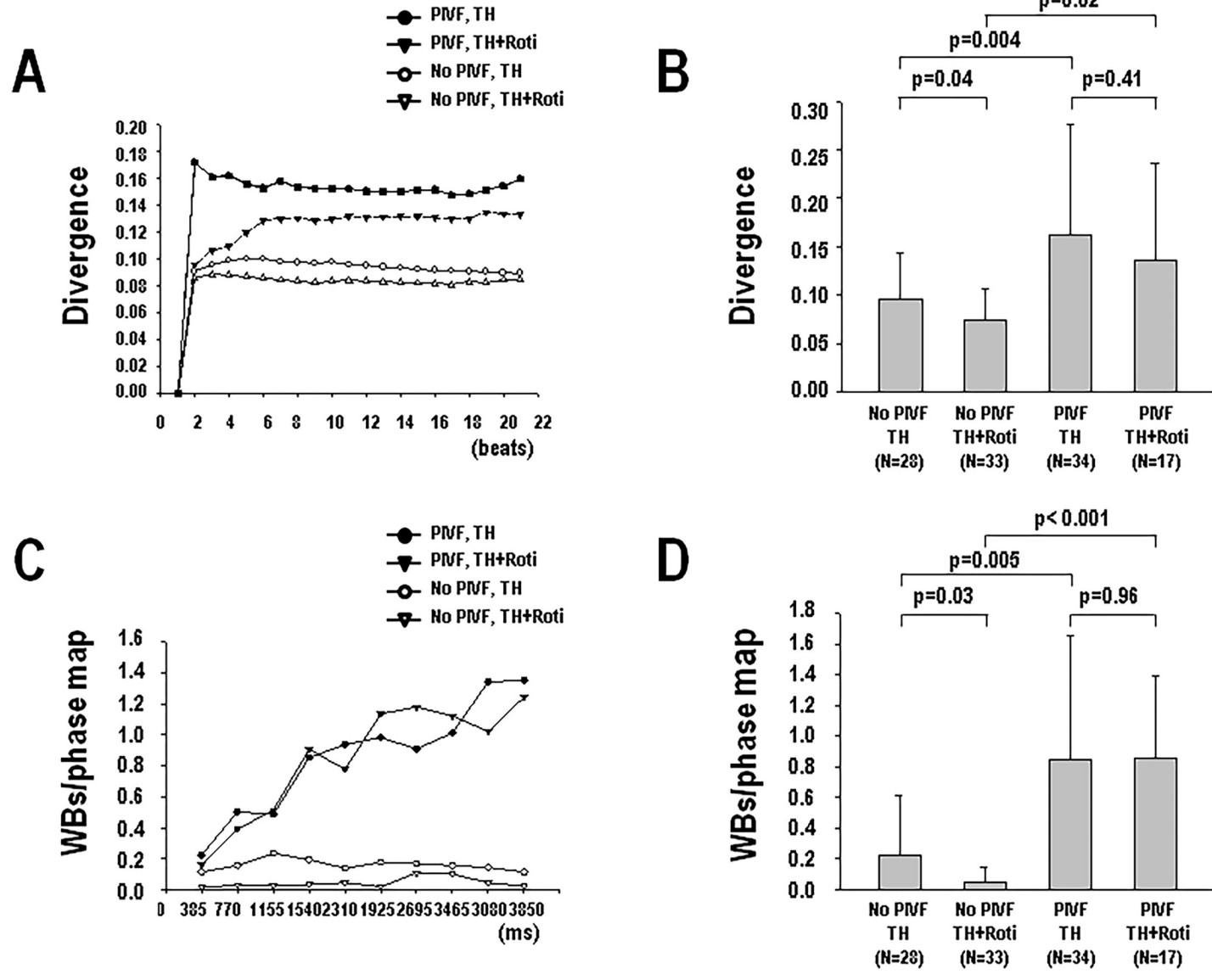

**Fig 5.** Beat-to-beat dynamic ventricular divergence (A, B) and epicardial wavebreaks (WBs) (C, D) during burst pacing in the pacing-induced ventricular fibrillation (PIVF) and non-PIVF episodes with and without rotigaptide (Roti) infusion during therapeutic hypothermia (TH, 30°C).

In the 112 pacing induction episodes, SDA was observed in 50 episodes while SCA was seen in 38 episodes. The divergence in the SDA episodes (0.11±0.06) was higher than that in the SCA episodes (0.08±0.04) (p = 0.03).

## Correlation between divergence and epicardial wavebreaks

Linear regression analysis showed a correlation existed between ventricular divergence and the numbers of epicardial wavebreaks during ventricular pacing at TH (p<0.001, $R^2 = 0.334$, Fig 8). This correlation between divergence and wavebreaks also exists in PIVF (p<0.001, $R^2 = 0.199$) and non-PIVF (p<0.001, $R^2 = 0.261$) episodes. In the episodes of PCL≤200, the correlation between divergence and wavebreak was significant (p<0.001, $R^2 = 0.313$). In the episodes

of PCL>200, the correlation between divergence and wavebreak was also significant (P<0.001, $R^2$ = 0.224).

## Different pacing sites on the divergence and VF inducibility

The divergence of the PIVF episodes was not different between RV (0.15±0.04) and LV (0.15±0.07) pacing (p = 0.97). Also, the divergence of the non-PIVF episodes was not different between RV (0.12±0.07) and LV (0.11±0.05) pacing (p = 0.60). The inducibility of VF was also similar before (p = 0.20) and after (p = 0.23) rotigaptide infusion between RV and LV pacing. These findings indicated that the divergences and VF inducibility in this model were not affected by changing the pacing sites.

## Extensive endocardial ablation on the divergence and VF inducibility

The divergence in the PIVF episodes was similar between the ablated (0.20±0.11, protocol IV) and non-ablation hearts (0.16±0.11, protocol I) (p = 0.38). The divergence in the non-PIVF

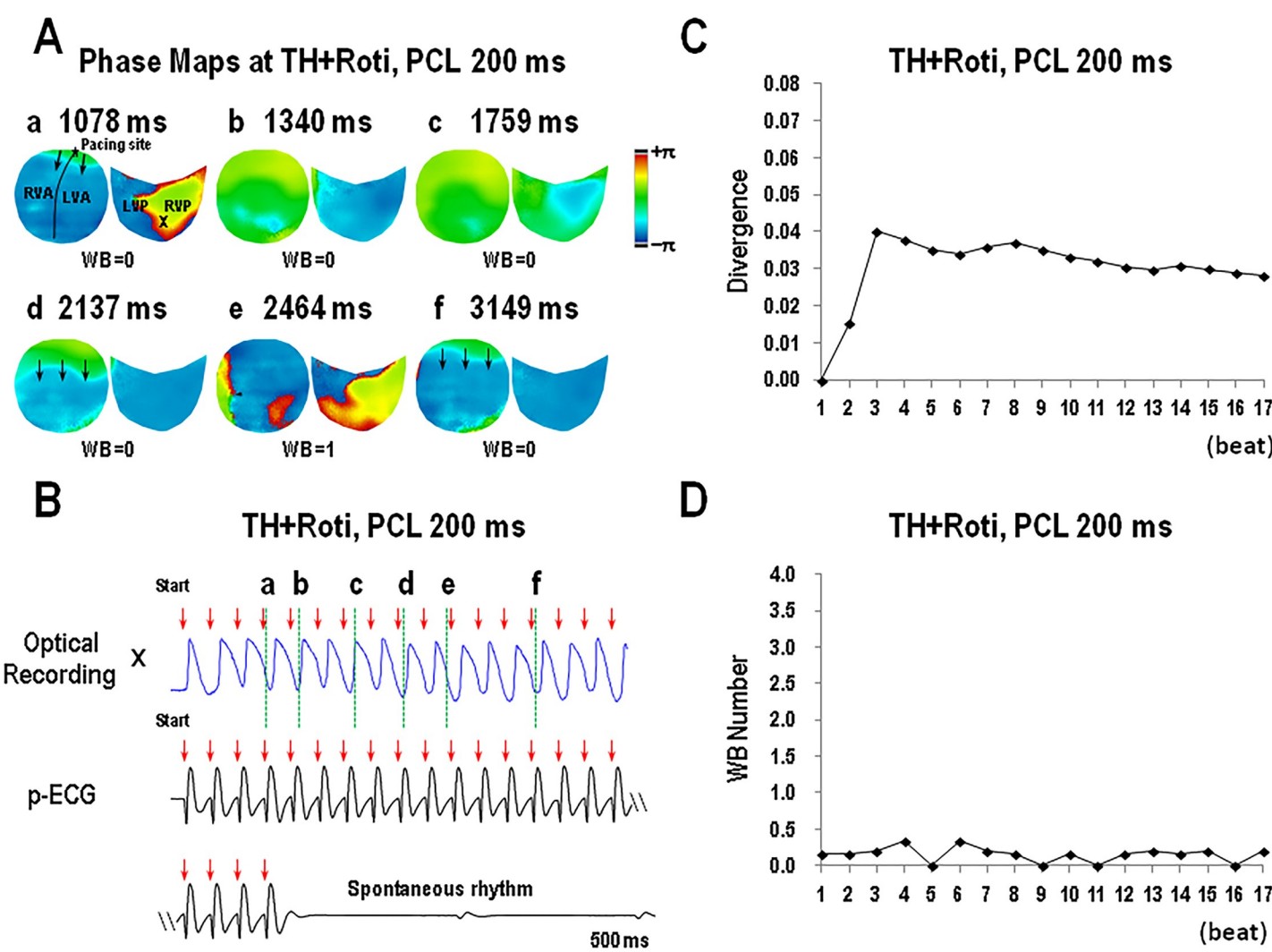

**Fig 6.** Simultaneous recordings of epicardial activations (A), pseudo-ECG (B), dynamic changes of divergence (C), and epicardial wavebreaks (D) at the start of burst ventricular pacing at pacing cycle length (PCL) 200 ms during therapeutic hypothermia (TH, 30°C) in an episode of low divergence (data is from heart #7). Note that the divergence increased to 0.040 at the 3rd beat and decreased to 0.028 at the 17th beat. VF was not inducible in this episode (B). Black triangles in panel A indicate points of wavebreaks (WBs).

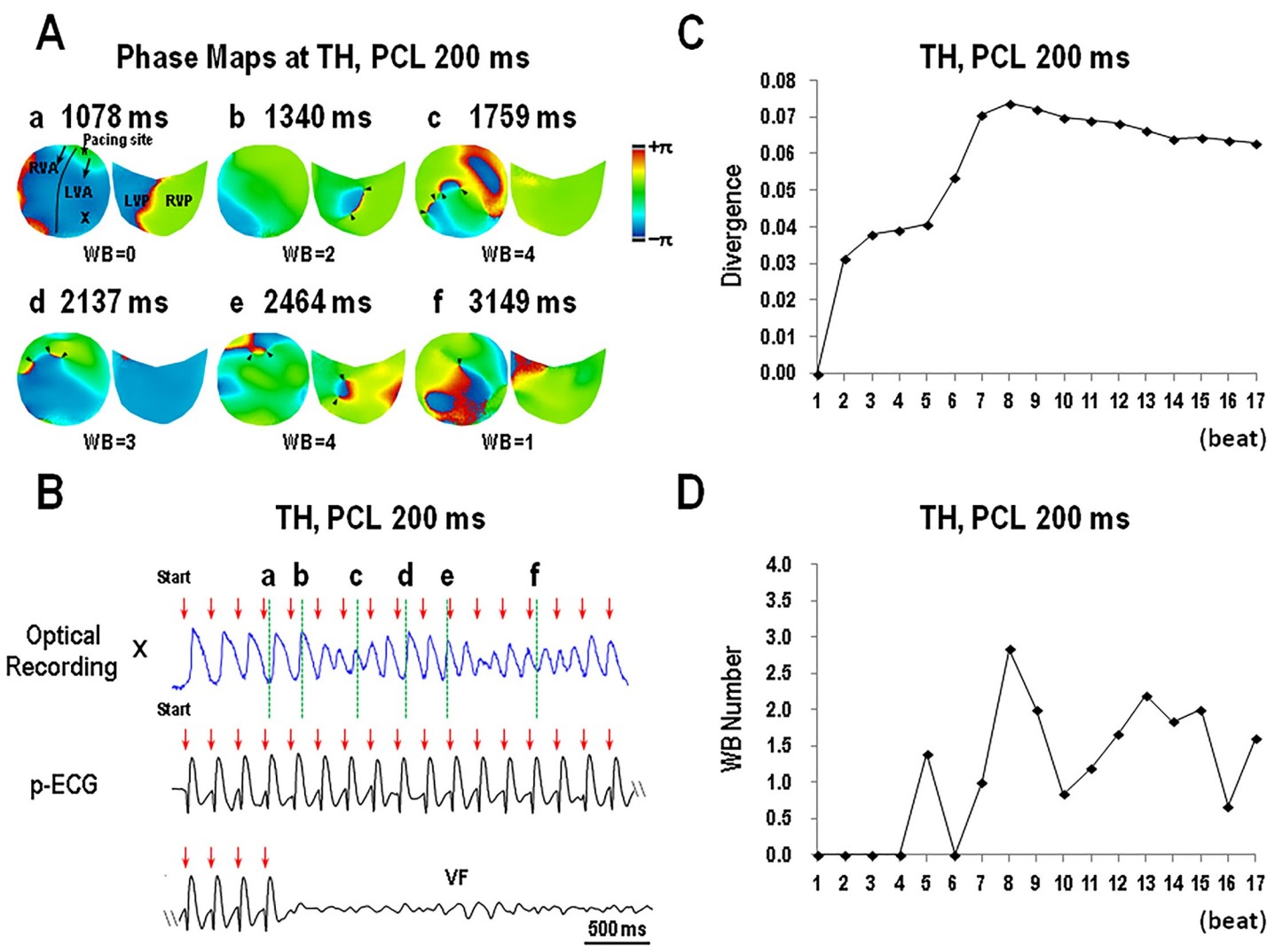

**Fig 7.** Simultaneous recordings of epicardial activations (A), pseudo-ECG (B), dynamic changes of divergence (C), and epicardial wavebreaks (D) at the start of burst ventricular pacing at pacing cycle length (PCL) 200 ms during therapeutic hypothermia (TH, 30°C) in an episode of high divergence (data is from heart #7). Note that the divergence increased to 0.031 at the 2nd beat and remained as high as 0.63 at the 17th beat. VF was inducible after stopping pacing (B). Black triangles in panel A indicate points of wavebreak (WBs).

episodes was similar between the ablated (0.09±0.02, protocol IV) and non-ablation hearts (0.09±0.07, protocol I) (p = 0.81). The wavebreaks in the PIVF episodes was similar between the ablated (0.52±0.25, protocol IV) and non-ablation hearts (0.85±0.71, protocol I) (p = 0.36). The wavebreaks in the non-PIVF episodes was similar between the ablated (0.10±0.12, protocol IV) and non-ablation hearts (0.11±0.26, protocol I) (p = 0.83). These findings indicated that endocardial sources might not have impact on the epicardial wavebreaks in this model. Rotigaptide decreased the VF inducibility from 50±14% to 0% (p = 0.04).

## Discussion

The major findings of this study are (1) The changes in ventricular divergence after rotigaptide infusion paralleled with the changes in epicardial wavebreaks in both the PIVF and non-PIVF episodes during TH. (2) Linear regression analysis showed a correlation between ventricular

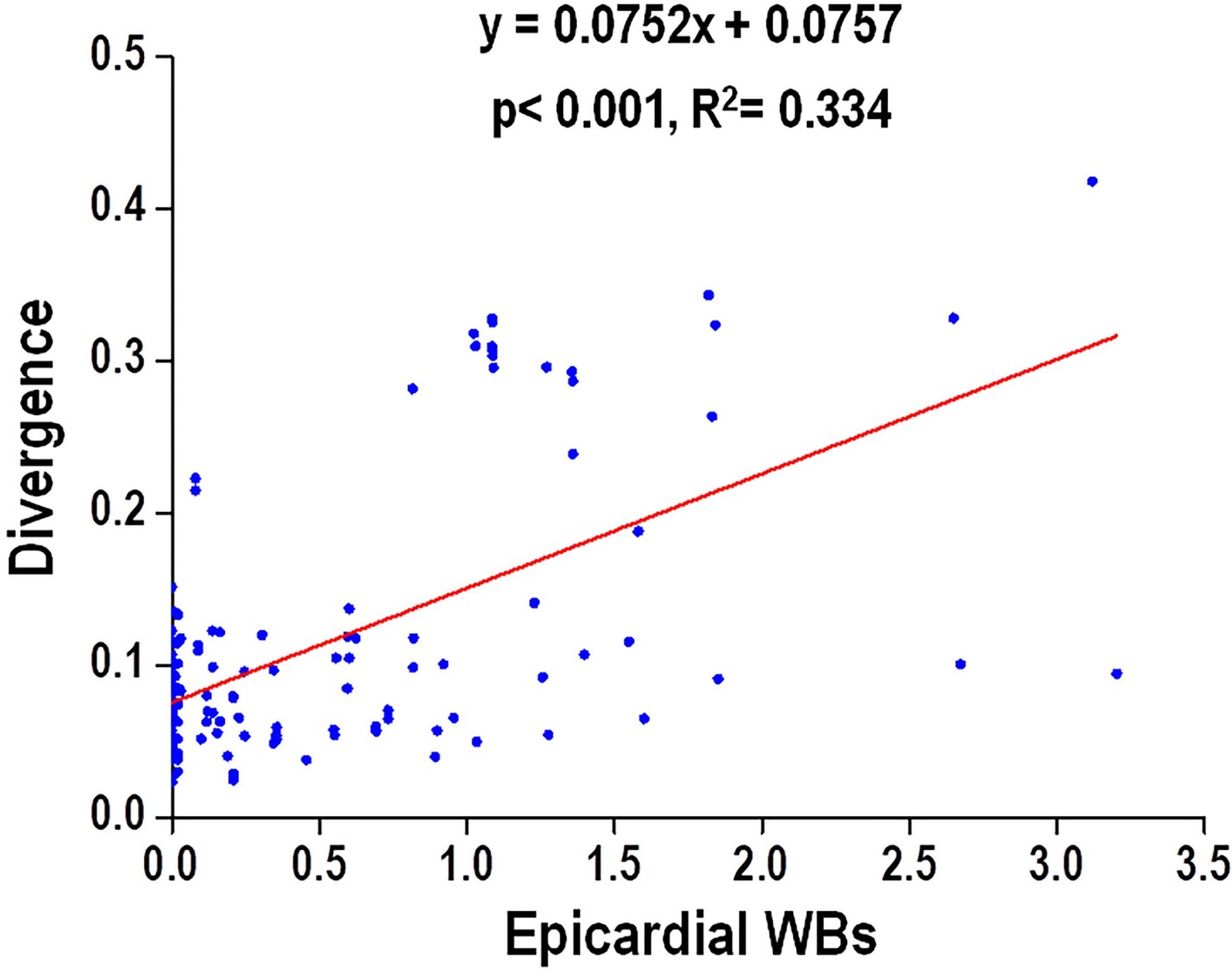

**Fig 8. Linear correlation of ventricular divergence versus epicardial wavebreaks (WBs) in all burst pacing episodes during therapeutic hypothermia (TH, 30˚C).**

divergence and the number of epicardial wavebreaks during ventricular pacing at TH. (3) Rotigaptide increased CV, and decreased both the ventricular divergence and the number of epicardial wavebreaks at ventricular pacing, and decreased the probability of PIVF during TH.

## Beat-to-beat morphological variation on arrhythmic susceptibility

Beat-to-beat morphological variation (divergence) in the duration and amplitude of ventricular electrograms been shown to predict VA and SCA among diverse patients with ischemic and non-ischemic cardiomyopathy [3]. Such subtle morphological variation might manifest as T wave alternans in ECG, which has been correlated with risk for VA and SCA [4]. By analyzing ambulatory ECG from patients, Shusterman et al found that the beat-to-beat morphological oscillations increased before the onset of spontaneous VF, indicating that these morphological variations might have a temporal relationship with the onset of VF [21]. In patients with an ICD, beat-to-beat morphological variation might even be recorded by ICD

leads before spontaneous VT and VF [5]. In patients with inadequate or irregular heart beat for divergence evaluation, PVS using burst ventricular pacing by electrodes or the atrial/ventricular leads of an implantable device has been found to be a reliable alternative to predict VA [6]. Although these findings indicated that these morphological variations might serve as a warn sign before VF, the mechanisms by which increased morphological variation lead to VA remains unclear.

Several mechanisms have been proposed to link the increased divergence with the occurrence of VA [22]. Increased heart rate could induce concordant alternans, which could progress to discordant alternans as the heart rate speed up and produce beat-to-beat morphological variation on ECG [23]. Discordant alternans causes steep gradients of repolarization that can provide the substrate for unidirectional functional block and the initiation VA [19]. Other study suggested that beat-to-beat morphological variation on ECG reflected increased heterogeneity in ventricular repolarization secondary to a steep repolarization restitution curve [22]. In this study, we observed a direct correlation between ventricular divergence and epicardial wavebreaks that leads to VF occurrence, providing another mechanism of increased morphological variation contributed to VF via epicardial wavebreak. Further studies are warrant to evaluate if this phenomenon still occurs in structurally diseased hearts.

## Ventricular divergence and epicardial wavebreaks on VF occurrence

TH has been shown to improve neurological outcomes in patients resuscitated from cardiac arrest [24]. Chorro et al reported that acute reduction of ventricular temperature to <20˚C might simplify VF activation patterns and terminated VF [25]. However, the temperature (<20˚C) they used is far below the therapeutic range of hypothermia. Harada et al found that TH at 30˚C in 2-dimensional rabbit ventricles enhanced wavebreaks and regeneration of new spiral waves during VF, facilitating VT and VF maintenance. This finding is consistent with the pro-arrhythmic model used in the current study [26]. In contrast, TH at 33˚C might increase the chance of spiral wave collision and VT/VF self-termination, indicating that 33˚C is a safe temperature for clinical use [26].

In this hypothermic (30˚C) and pro-arrhythmic rabbit heart model, we found that the divergence had changed in parallel with the changes in epicardial wavebreaks during TH. To establish their correlation, we used rotigaptide treatment to reduce epicardial wavebreaks as in our previous study [11], and found that the divergence had decreased as well (Fig 4B and 4D). The infusion of rotigaptide also simultaneously reduced both divergence (Fig 5B) and wavebreaks (Fig 5D) in episodes of non-PIVF. Linear regression analysis showed that the divergence and epicardial wavebreak numbers were correlated. These findings suggest that ventricular divergence on ECG could be a predictor of epicardial wavebreaks during PVS.

In the present study, episodes of PIVF showed a higher divergence and more epicardial wavebreaks when compared to episodes of non-PIVF. Infusion of rotigaptide not only reduced both divergence and epicardial wavebreaks, but also decreased the probability of PIVF, indicating that reducing both divergence and epicardial wavebreaks could lead to less VF. In multiple wavelet hypothesis, wavebreaks resulting from collisions of propagation mother wavelets might generate daughter wavelets in a self-sustaining turbulent process to maintain VF [27]. Using optical mapping techniques, PS found on phase maps is the area of ambiguous activation state which underlies the formation of rotor and wavebreaks, and serves as a source of VF [10]. Taken together, the increase in divergence during ventricular pacing could indicate multiple epicardial wavebreaks, which is essential for VF maintenance. Infusion of rotigaptide lowered the number of epicardial wavebreaks, and reduced the probability of PIVF.

Because epicardial wavebreaks cannot be observed in clinical practice, patients with high ventricular divergence during PVS should be carefully evaluated in order to identify underlying instability of myocardial substrate. Recently, a 3D multi-electrode vest which can gather cardiac electrophysiological data from the body surface has become widely used [28]. With this technology, one would be able to obtain bi-atrial and bi-ventricular, 3-D cardiac activation maps during PVS, and thus providing a direct correlation with divergence and epicardial wavebreaks in the future.

## Anti-arrhythmic mechanisms of rotigaptide during TH

Rotigaptide is an anti-arrhythmic peptide which enhances gap junction intercellular conductance between cardiomyocytes, without changing the membrane conductance [29]. Kjolbye et al reported that rotigaptide hinders the onset of arrhythmogenic SDA, and protects ventricular tachyarrhythmia in an ischemic guinea pig heart model [30]. We also found that rotigaptide could enhance CV, prevent SDA, and decrease VF occurrence in hypothermic (30°C) isolated rabbit hearts [11]. In the present study, we discovered that ventricular beat-to-beat variation (divergence) during burst pacing was decreased by rotigaptide. Additionally, ventricular divergence in SDA episodes is higher than that in SCA episodes. It is possible that high ventricular divergence prior to rotigaptide infusion during TH is related to SDA in a minor extent, and that rotigaptide increased CV (Fig 1), suppressed the SDA and reduced the divergence [11]. This speculation is supported by the observation that SDA causes epicardial wavebreaks and VF occurrence in guinea pig hearts [31]. Although changing the APD restitution curve may predispose to wavebreak and VF during pacing [32], our data showed that rotigaptide did not affect APD restitution during TH (Fig 1) [11]. Therefore, the suppression of divergence by rotigaptide could be associated with a reduced SDA, wavebreak number, and less inducible VF. Further studies are still needed to test this hypothesis.

## Clinical implications

Beat-to-beat morphological variation during PVS is a robust indicator for epicardial wavebreak, which may lead to VA. An ICD has been widely used to effectively prevent SCA and prolong patient survival in primary and secondary prevention clinical trials [33]. To minimizing any unnecessary ICD shock, one approach is to early identify the vulnerable myocardial substrate at high risk of VA before VA occurrence. Given the prevalence of ventricular pacing in ICD patients, increased beat-to-beat morphological variation during ventricular pacing might be a warn sign of epicardial wavebreaks and recurrent VA. Delivery of essential anti-arrhythmic therapy or revascularization before VA occurrence might be of great value to prevent electrical storm. In patients resuscitated from SCA who are undergoing TH, rotigaptide infusion may decrease ventricular divergence and epicardial wavebreaks, thus reducing the risk for recurrent VA.

## Limitations

The present study had some limitations. Firstly, we used this hypothermic model to elucidate the correlation between ventricular divergence and epicardial wavebreak during PVS in normal ventricles. Whether our experimental conclusions can be extended to other experimental models (ischemic or heart failure models) remains to be verified. Using computer simulation would also be beneficial in controlling experimental conditions and prove our hypothesis in the future. Secondly, our optical mapping data included PSs recorded from the epicardial surface. One cannot exclude the possibility that intramural wavebreaks were altered by the use of rotigaptide, and therefore had contributed to changes in divergence. Thirdly, the effects of

rotigaptide on divergence and wavebreaks during TH were evaluated exclusively after short treatments of 20 minutes. Whether longer durations of rotigaptide infusion (i.e., 12–24 hours, as compliant with clinical guidelines) could similarly reduce divergence and wavebreak remains to be determined. Fourthly, we chose the maximal effective concentration (300 nM) to evaluate the divergence-wavebreaks relationship in normal ventricles. Whether the positive correlation observed in this study could be applied to rotigaptide at other concentration or in the presence of heterogeneous structures remains to be explored. Although linear regression showed a correlation between ventricular divergence and the numbers of epicardial wavebreaks, the low R square value indicated that divergence along might not strongly predicted the wavebreaks numbers. Interpreting this data should be with caution. Finally, despite the electrophysiological properties including APD and CV were not changed with rotigaptide or vehicle infusion during TH, we could not completely exclude the possibility that protein functioning and metabolisms have been altered by TH per se. Further control study at 37˚C is warrant.

## Conclusions

Ventricular beat-to-beat morphological variation evaluated as divergence was correlated with, and might be predictive of, epicardial wavebreak during ventricular pacing at TH. Treatment with rotigaptide reduced both the ventricular divergence and epicardial wavebreak at ventricular pacing, and decreased the probability of PIVF during TH. Enhancing cell-to-cell coupling could be a novel approach toward improving the stability of myocardial substrate during TH.

## Supporting information

**S1 Fig. Experimental time of the study protocol.**
(TIF)

**S1 Table. Pacing cycle lengths and results of each attempt before and after rotigaptide treatment during hypothermia (30˚C).**
(DOCX)

**S1 Movie. Example of pacing induction of VF at PCL 200 ms during hypothermia.**
(AVI)

**S1 Data. Divergence raw data during hypothermia.**
(XLSX)

## Acknowledgments

We thank Ya-Wen Hsu, Hung-De Yuan, and Shane Su for their technical assistance. Rotigaptide was supplied by Zealand Pharma, Glostrup, Denmark.

## Author Contributions

**Data curation:** Yu-Cheng Hsieh, Wan-Hsin Hsieh, Men-Tzung Lo.

**Formal analysis:** Yu-Cheng Hsieh, Wan-Hsin Hsieh, Cheng-Hung Li, Ying-Chieh Liao, Jiunn-Cherng Lin, Chi-Jen Weng, Men-Tzung Lo, Ta-Chuan Tuan, Yenn-Jiang Lin, Wei-Wen Lin.

**Investigation:** Yenn-Jiang Lin.

**Methodology:** Shien-Fong Lin, Jin-Long Huang, Tsu-Juey Wu, Shih-Ann Chen.

**Project administration:** Wan-Hsin Hsieh.

**Resources:** Ketil Haugan, Bjarne D. Larsen.

**Writing – original draft:** Yu-Cheng Hsieh.

**Writing – review & editing:** Hung-I Yeh, Ketil Haugan, Tsu-Juey Wu, Shih-Ann Chen.

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
