## [Decision Letter · Decision Letter 0]

8 Nov 2019

PONE-D-19-26993

Ventricular Divergence Correlates with Epicardial Wavebreaks and Predicts Ventricular Arrhythmia in Isolated Rabbit Hearts during Therapeutic Hypothermia

PLOS ONE

Dear Dr. Hsieh,

Thank you for submitting your manuscript to PLOS ONE. After careful consideration, we feel that it has merit but does not fully meet PLOS ONE’s publication criteria as it currently stands. Therefore, we invite you to submit a revised version of the manuscript. As you will gather from the reviews, the referees identified a number of substantive conceptual and methodological problems that must be addressed.

We would appreciate receiving your revised manuscript before January 10th, 2020. To enhance the reproducibility of your results, we recommend that if applicable you deposit your laboratory protocols in protocols.io, where a protocol can be assigned its own identifier (DOI) such that it can be cited independently in the future. For instructions see: http://journals.plos.org/plosone/s/submission-guidelines#loc-laboratory-protocols

We look forward to receiving your revised manuscript.

Kind regards,

Manuel Zarzoso, Ph.D.

Academic Editor

PLOS ONE

Journal Requirements:

2. To comply with PLOS ONE submissions requirements, in your Methods section, please provide additional information on the animal research and ensure you have included details on (1) methods of sacrifice, (2) methods of anesthesia and/or analgesia, and (3) efforts to alleviate suffering.

This study was supported by grants from the National Science Council (104-2314-B-367-001 and 105-2314-B-075A-016 -MY3), Taipei, Taiwan; and Taichung Veterans General Hospital (TCVGH-1033105C, TCVGH-1043109C, TCVGH-1053108C, VGHUST104-G5-2-2, VGHUST107-G5-1-3, TCVGH-VHCY1068606, and TCVGH-VHCY1078603), Taichung, Taiwan.

We note that one or more of the authors are employed by a commercial company: Zealand Pharma A/S

Reviewers' comments:

Reviewer's Responses to Questions

**Comments to the Author**

1. Is the manuscript technically sound, and do the data support the conclusions?

Reviewer #1: Partly

Reviewer #2: No

Reviewer #3: Partly

2. Has the statistical analysis been performed appropriately and rigorously? 

Reviewer #1: Yes

Reviewer #2: Yes

Reviewer #3: N/A

3. Have the authors made all data underlying the findings in their manuscript fully available?

Reviewer #1: Yes

Reviewer #2: Yes

Reviewer #3: Yes

4. Is the manuscript presented in an intelligible fashion and written in standard English?

Reviewer #1: Yes

Reviewer #2: Yes

Reviewer #3: Yes

5. Review Comments to the Author

Reviewer #1: In this manuscript, the authors investigated the mechanism underlying anti-tachycardia pacing (ATP) failure and how ATP can initiate ventricular fibrillation. Using rabbit hearts, the authors conclude that high ventricular divergence at ECG during ATP is associated with more epicardial wavebreaks, which then lead to initiation of pacing (probably ATP) induced ventricular fibrillation.

Comments from the reviewer:

Major:

1. In this study, the authors applied therapeutic hypothermia (TH) to create arrhythmogenic substrates, which does not physiologically exist during ATP in real life. The experimental design significantly limited the type of VF (regarding to its underlying mechanism) that ATP would initiate, which may not be the case in patients. Therefore, the conclusion and the significance of this study is overstated.

2. ATP, in real life, only applies to patients during VT, which is not the experimental design in this study. Again, the conclusion from this experimental design cannot deduct the conclusion which can transfer to patients. Especially, in real life, VT in patients may have multiple mechanisms.

3. As the authors mentioned, APT failure is associated with the pacing site and conduction time. However, the pacing site factor was not considered in this study. Does changing pacing site for ATP really can increase the success rate? This is easy to conduct as the optical mapping can directly guide the authors to find the VT reentry circuit.

4. The authors applied baseline test and drug test in the same rabbit heart in their protocol during TH. Whether TH will affect the heart during long time period which changes the protein functioning, metabolisms, electrophysiological properties was not tested in this study. The authors needs to prove that the VF inducability is consistent between baseline testing (30min during TH) and drug application (more than 1 hours during TH).

5. The authors only applied epicardial optical mapping, and conclude that the epicardial wavebreaks are the mechanisms for ATP initiating VF. This is not accurate and solid, the authors need more experiments to prove that it is not endocardium wavebreak and propagate to epicardium during the optical mapping.

Minor:

5. Why the authors use such long (3.85ms) exposure time in optical mapping experiments? Will this enough to capture fast wave breaks? There are multiple papers reporting optical mapping in rabbit ventricle with 1k and even more sampling frequency with sound signal to noise ratio.

6. Please clarify how 112 optical /p-ECG recordings are collected, at what time, right after pacing applied or wait for some time? Are they all the same regarding to the arrhythmogenesis? Some vulnerable time window may have larger arrhythmia possibility.

7. Figure1A looks weird, the conduction velocity looks much smaller than the ones showed in Figure 1B.

Reviewer #2: The submitted manuscript examined the susceptibility for pacing induced VF (PIVF) in isolated rabbit hearts under conditions of hypothermia (TH) and TH + Rotigaptide. TH led to a reduction in conduction velocity and prolongation of the APD, compared to baseline (37C.) The gap junction modifier Rotigaptide increased the conduction velocity, and reduced the susceptibility for PIVF. Similarly, Rotigaptide led to reduced ECG divergence and wavebreaks. There was a correlation between divergence and wavebreaks.

While the results are very interesting, and increased divergence and wavebreaks were associated with PIVF, it is difficult to assess the reliability of the results, as there may be other underlying confounders. Clarification of the conditions for the analyses and whether the results are from all animals (lumped together) or from specific animals that did vs did not develop PIVF, will help indicate whether divergence and wavebreaks may be markers for PIVF. While the introduction, discussion, goals of the study, and conclusions are about which patients will respond to anti-tachycardia pacing, the present results are not studying the success of anti-tachycardia pacing, but rather the susceptibility for PIVF. Therefore, the results do not support the conclusions.

ATP is performed to stop an arrhythmia, and not performed during sinus rhythm or quiescence. Thus, as this study is assessing the susceptibility and ECG dynamics (wavebreak & divergence) for PIVF, this study did not investigate or provide results to indicate which patients will respond successfully to ATP.

As there were inter-animal differences in the pacing frequency and number of pacing episodes, results need to be more standardized, without these underlying confounders. For example:

-Was PIVF achieved in all rabbits? If not, then do divergence and wavebreak serve as markers for animals that will develop PIVF or PIVF episodes about to arise? If not every rabbit developed PIVF, please present the number or percent of rabbits that developed PIVF (not episodes.) When presenting results on wavebreak and divergence, clarify whether the results are from PIVF vs. no PIVF animals. For example, are no PIVF episodes only from animals that did not ever develop PIVF, or just an individual pacing episode that didn’t result in PIVF? If it is the later then the results in the study do not indicate which animals will develop PIVF (or success of ATP), but rather a pacing protocol that will not lead to PIVF.

-As each animal was exposed to 5-10 episodes of PIVF, results should be presented per rabbit, as some rabbit results (i.e., those with 10 pacing episodes) artificially skew the data compared to those with only 5 pacing episodes.

-Fig. 2, how many pacing episodes came from each of the 10 animals? Was there a relation between PIVF probability and pacing frequency?

-Fig. 3 shows snapshots of high vs. low divergence, but these are from different animals (rabbit#10 vs. rabbit#7) and possibly different pacing frequencies. Thus, it is unclear if the decrease in divergence was due to Rotigaptide, pacing frequency, animal, or another confounder.

-Fig. 4 shows 112 pacing episodes (PIVF n=51, no PIVF n=61), but it is unclear how many came from each animal, and what were the pacing frequencies. Thus, it is not clear if differences in divergence and wavebreak is purely due to PIVF vs. no PIVF. Similarly, there is not a 1-to-1 relation of pacing episodes at TH vs. TH+Roti. Thus, it is unclear if Rotigaptide reduced divergence and wavebreak, or it was due to another confounder. Please provide percent change at TH+Roti compared to TH in the same animal and pacing frequency.

-Fig. 6 and 7 – If you want to indicate differences in waveform, divergence, and wavebreak in the presence (Fig. 6) and absence of Rotigaptide (Fig 7), then they need to be performed under the same conditions (i.e., same animal and pacing frequency), and not pacing at 220 ms vs. 280 ms.

-Fig. 8 indicates the relationship between wavebreak and divergence, which is to be expected. It would be interesting to assess the correlation between wavebreak and divergence points from animals and episodes that only did vs. did not develop PIVF. Importantly, with proper analyses, these results could potentially indicate a marker for PIVF.

Reviewer #3: Hereby the authors describe a study trying to shed light into the functional mechanisms of ATP therapy failure when high beat-to-beat morphological ‘dispersion/divergence’ on ventricular electrograms is present. Their hypothesis is that epicardial wavebreaks on the heart surface correlate to pacing induced morphological heterogeneity and predicts higher inducibility of ventricular arrhythmia. To test their hypothesis the investigators propose a study using optical mapping on the isolated Langendorff-perfused rabbit heart model, defining a protocol to test VF inducibility after hypothermia (30ºC, 30’) and, subsequent, Rotigaptide treatment (300nM, 20’). Despite describing their major limitations a critical concern is that the model and protocol presented by the authors may present bias not allowing them to support their conclusions and clinical implications; and, further, extrapolate to conditions with real ischemic or other substrate scenarios where ATP therapy success or failure is critical.

Criticisms:

- The authors claim that divergence (morphological variation of electrical activity during pacing) correlates significantly to epicardial wavebreaks in the surface, however, it is clear from their adjusted R-squared goodness of fit, and figure 8, that the relationship between divergence and epicardial wavebreaks seems to poorly predict the data. Thus the authors should be cautious with their conclusions and discuss the implications carefully even if their claim is only that a significant relationship may exist, and being just one regressor, may have some average predictive power.

- Even though the authors have published the concept of ‘divergence of an electrical signal’ before, does not seem sound when speaking of morphological correlation or dispersion of signal patterns. Divergence is a concept used to represent the volume densities of the outward flux of vector fields when applied in the context of scalar fields such electrical or optical fields. Therefore, it may accurately be used to describe singularities in the heart surface, yet confusing when applied to the pseudo-ecg or ekg signals, or any other signal parameter dispersion. Instead, a classic correlation or entropic measure, deviation, or even turbulence from each beat could also represent significantly the instability of activation during susceptible induction.

- Different studies (including the authors) have published before how arrhymogenic substrates may be favored by 30º-hypothermia, facilitating VF formation, alternans and wavebreaks. However, other authors, such as, Harada et all AJP Heart Circ Physiol 2008, or, Chorro FJ et al (AJP Heart Circ Physiol 2002), have provided strong evidence suggesting the antiarrhythmic therapeutic potential of moderate hypothermia, by also promoting self-termination of VT/VF. Please discuss. The authors only found a mild reduction in wavebreak formation under Rotigaptide treatment up to the onset of induction. Which was the lifetime of the wavebreaks (as measured) under each condition? Were they monitored only coupled with pacing beats? Was the duration or amount of wavebreaks directly related to the induction frequency or degree of morphological dispersion? Please discuss.

- The Rotigaptide dose used in the study is quite high. What are the dose-response effects of Rotigaptide on induction mechanisms (divergence-wavebreaks)? Does it have antiarrythmic properties at higher concentrations? How is it modified in presence of anatomical substrate in presence of heterogenous tissue, for example, such as the presence of RF lesions, which may be more relevant to ATP therapy testing?

- Including computer simulations may benefit the study testing different scenarios to prove the author’s hypotheses without intrinsic experimental limitations, and further, inexpensively. For example, the authors could test different pacing conditions including obstacles, regional differences in conductivities, distances to the obstacle, etc and controlling the experimental conditions. Have the authors determined before which cut-off ‘morphological divergence/dispersion’ values results in true ATP-failure or greater VT/VF inducibility probability?

- A major concern is the protocol. Is the heart submerged during the hypothermia protocol? Did you monitor the temperature during the protocol in the bath and heart surface? Why didn’t the authors include a control group (same time-stamped registers with no treatment) to rule out experimental constraints over time?

- While the idea behind the study may potentially be interesting, ‘to correlate beat-to-beat morphological dispersion during ATP failure to actual heart surface dynamics, and specifically to wavebreaks, under hypothermia’, the longitudinal protocol design, testing inducibility by burst pacing (only accounting to acute temperature-related modifications) make the results presented very limited. Severe hypothermia as shown by the authors before may be arrhythmogenic, which means that under fast pacing re-excitation and unidirectional block may appear leading under such substrate conditions to sustained or short-lived rotating activity. The question would be how early the divergence measure is sensitive and starts to accurately correlate to short-lived wavebreaks (when ventricular arrhythmia is not yet initiated). Otherwise, the relationship is well known already. Scenarios shown in figures 6 and 7 could shed some light in to it, however it is not clear in B weather optical and electrical signals temporarily match, and if so, how the extra beats 16-17-18 are related? They are not represented in C-D beat-to-beat single measures. Animated movies or videos illustrating the process under each condition would help interpretation and readability.

- In figure 8: Please consider incorporating the association under each specific condition. Is the presumed significant correlation frequency of induction-dependent?

- The authors use epicardial wavebreaks as a key measure of arrhythmia susceptibility or arrhythmia promoting and driving sources; yet wavebreaks measured as singularities by visual inspection, may represent sustained reentry after unidirectional block, simple to complex wavefront collisions from distant driving activity or even fibrillatory conduction. First, the methodology to track those points is not detailed enough, are detected every frame? In that case, the authors could provide detail information on the spatiotemporal properties comparing PIVF and No PIVF, both before and after Roti treatment. Which is the mechanism of initiation under each of the experimental conditions tested? Could it be possible that the mechanism is frequency of induction dependent, and therefore, better predicted or not by morphological variation on the pacing pseudo-EG?

Minor comments:

- How long lasted the whole protocol? Please provide a clarifying figure.

- Through the study the authors use APD70 as a measure of APD. At least APD80 should be measured in optical mapping studies.

- How were the pseudo-ecg signals processed before applying the proposed ‘divergence’ equation?

- Methods section should be revised for innacuracies and, potentially, partially moved to a supplementary section referencing previous studies when possible.

- Figure2. Please complete information including SD/SE bards. Details on how VF episodes were considered should be included in the methods section, even include an episode in figure 2. Furthermore, which were the entry frequency and mean VF dominant frequency under each condition across the study sample.

- Being only n=10 animals, including the individual data onto the bars would benefit the interpretation of the data.

- In figure 3, different PCLs are used so the interpretation, ¿why? How different were the baseline spontaneous frequencies at the beginning of the protocol and before rotigaptide treatment. ¿How long was the protocol?

- In figure 4, WB/phase map are compared when pacing induced VF, and, under Roti treatment. How were WB defined? Are they normalized? Why mean values are less than 1? Perhaps singularities should be measured automatically in each map, per temporal and physical unit in the whole field of view.

- Pacing cycle lengths used in the figures 6-7 are not reflected in the CV restitution curves ¿why? Frequencies used for the burst pacing protocol are not clearly specified in the methods section.

- Differences in ‘divergence’ o measured ‘WB’ values seem rather small over time and the sample size limited for some comparisons. Which were your expected effect size and statistical power to estimate that sample size?

6. PLOS authors have the option to publish the peer review history of their article (what does this mean?). If published, this will include your full peer review and any attached files.

Reviewer #1: No

Reviewer #2: No

Reviewer #3: No

---

## [Author Response · Author response to Decision Letter 0]

9 Jan 2020

MS Number: PONE-D-19-26993R1

Responses to Editor

Comments to the Author

Journal Requirements:

Response:

We have adjusted our manuscript to meet PLOS ONE's style requirements.

2. To comply with PLOS ONE submissions requirements, in your Methods section, please provide additional information on the animal research and ensure you have included details on (1) methods of sacrifice, (2) methods of anesthesia and/or analgesia, and (3) efforts to alleviate suffering.

Response:

In the method section, we have addressed this point as below “Before anesthetization, ketamine (10 mg/kg) was injected intramuscularly to calm the animals. After 10–15 min, the rabbits were intravenously injected with heparin (1,000 units) and anesthetized with sodium pentobarbital (35 mg/kg) via the marginal ear vein. After a median sternotomy, the hearts were rapidly excised.” (p 4, para 3 to p 5, para 1)

This study was supported by grants from the National Science Council (104-2314-B-367-001 and 105-2314-B-075A-016 -MY3), Taipei, Taiwan; and Taichung Veterans General Hospital (TCVGH-1033105C, TCVGH-1043109C, TCVGH-1053108C, VGHUST104-G5-2-2, VGHUST107-G5-1-3, TCVGH-VHCY1068606, and TCVGH-VHCY1078603), Taichung, Taiwan.

We note that one or more of the authors are employed by a commercial company: Zealand Pharma A/S

Response:

We have updated this author’s role (Dr. Bjarne D. Larsen) in the Author Contributions section of the online submission form.

Response:

We have amended our Funding Statement as below: “Zealand Pharma A/S provided support in the form of salaries for author [BDL] and rotigaptide, but did not have any additional role in the study design, data collection and analysis, decision to publish, or preparation of the manuscript.”

2. Please also provide an updated Competing Interests Statement declaring this commercial affiliation along with any other relevant declarations relating to employment, consultancy, patents, products in development, or marketed products, etc. Within your Competing Interests Statement, please confirm that this commercial affiliation does not alter your adherence to all PLOS ONE policies on sharing data and materials by including the following statement: "This does not alter our adherence to PLOS ONE policies on sharing data and materials.” (as detailed online in our guide for authors http://journals.plos.org/plosone/s/competing-interests) . If this adherence statement is not accurate and there are restrictions on sharing of data and/or materials, please state these. Please note that we cannot proceed with consideration of your article until this information has been declared.

Response:

In the “Competing Interests Statement”, we have stated that “Zealand Pharma A/S provided support in the form of salaries for author [BDL] and rotigaptide. This does not alter our adherence to PLOS ONE policies on sharing data and materials.”

Response:

We have included an updated “Funding Statement” and “Competing Interests Statement” in the cover letter.

 

MS Number: PONE-D-19-26993R1

Responses to Reviewer #1

Comments to the Author

In this manuscript, the authors investigated the mechanism underlying anti-tachycardia pacing (ATP) failure and how ATP can initiate ventricular fibrillation. Using rabbit hearts, the authors conclude that high ventricular divergence at ECG during ATP is associated with more epicardial wavebreaks, which then lead to initiation of pacing (probably ATP) induced ventricular fibrillation.

Major:

1. In this study, the authors applied therapeutic hypothermia (TH) to create arrhythmogenic substrates, which does not physiologically exist during ATP in real life. The experimental design significantly limited the type of VF (regarding to its underlying mechanism) that ATP would initiate, which may not be the case in patients. Therefore, the conclusion and the significance of this study is overstated.

Response:

We thank the reviewer for this insightful comment. Since ATP was only delivered to a pre-existing VT, the current experimental design was not compatible with ATP therapy. Therefore, we used programmed ventricular stimulation (PVS) instead of ATP to describe the burst ventricular pacing protocol, and to correlate the beat-to-beat morphological variation (divergence) with epicardial wavebreaks during TH. Accordingly, we have revised the introduction, discussion, and clinical implication sections to avoid misunderstanding or over-statement. (p 3-4; p 15, para 2 to p 16, para 1; p 19, para1) 

2. ATP, in real life, only applies to patients during VT, which is not the experimental design in this study. Again, the conclusion from this experimental design cannot deduct the conclusion which can transfer to patients. Especially, in real life, VT in patients may have multiple mechanisms.

Response:

We thank the reviewer for this insightful comment. Since ATP was only delivered to a pre-existing VT, the current experimental design was not compatible with ATP therapy. Therefore, we used programmed ventricular stimulation (PVS) instead of ATP to describe the burst ventricular pacing protocol, and to correlate the beat-to-beat morphological variation (divergence) with epicardial wavebreaks during TH. Accordingly, we have revised the introduction, discussion, and clinical implication sections to avoid over-statement. (p 3-4; p 15, para 2 to p 16, para 1; p 19, para1) 

3. As the authors mentioned, APT failure is associated with the pacing site and conduction time. However, the pacing site factor was not considered in this study. Does changing pacing site for ATP really can increase the success rate? This is easy to conduct as the optical mapping can directly guide the authors to find the VT reentry circuit.

Response:

We have performed an additional protocol (protocol III) to evaluate the divergence between two different pacing sites (RV and LV). (p 7, para 3) 

We observed that the divergence of PIVF (p=0.74) and non-PIVF (p=0.99) episodes was not different between RV pacing and LV pacing. The inducibility of VF was also similar before (p=0.37) and after (p=0.20) rotigaptide infusion between RV and LV pacing. These findings indicated that the divergences and VF inducibility in this experiment were not affected by changing the pacing sites. (p 14, para 2) 

4. The authors applied baseline test and drug test in the same rabbit heart in their protocol during TH. Whether TH will affect the heart during long time period which changes the protein functioning, metabolisms, electrophysiological properties was not tested in this study. The authors need to prove that the VF inducibility is consistent between baseline testing (30 min during TH) and drug application (more than 1 hour during TH).

Response:

To exclude the time effect on the consistency of VF inducibility test, we have added a new protocol (protocol II) using saline instead of rotigaptide in 4 hearts to serve as the control group. (p 7, para 2) 

We found that the VF inducibility was similar before and after saline infusion during TH. Also, we observed that the APD and CV were indifferent before and after saline infusion in the control group. (p 11, para 2)

As the reviewer mentioned, we did not evaluate the protein functioning and metabolisms in the rotigaptide and groups. We have included this part in the limitation section as “Although the electrophysiological properties including APD and CV were not changed with rotigaptide or vehicle infusion during TH, we could not completely exclude the possibility that protein functioning and metabolisms have been altered by TH per se. Further control study at 37°C is warrant.” (p 20, para 1)

5. The authors only applied epicardial optical mapping, and conclude that the epicardial wavebreaks are the mechanisms for ATP initiating VF. This is not accurate and solid, the authors need more experiments to prove that it is not endocardium wavebreak and propagate to epicardium during the optical mapping.

Response:

We have added a new protocol (protocol IV) to evaluate the impacts of endocardial sources on the epicardial wavebreaks by extensive endocardial ablation with Lugol’s solution. (p 7, para 4) We observed that: 

The divergence in the PIVF episodes was similar between the ablated (0.20±0.11, protocol IV) and non-ablation hearts (0.16±0.11, protocol I) (p=0.38). The divergence in the non-PIVF episodes was similar between the ablated (0.09±0.02, protocol IV) and non-ablation hearts (0.09±0.07, protocol I) (p=0.81). The wavebreaks in the PIVF episodes was similar between the ablated (0.52±0.25, protocol IV) and non-ablation hearts (0.85±0.71, protocol I) (p=0.36). The wavebreaks in the non-PIVF episodes was similar between the ablated (0.10±0.12, protocol IV) and non-ablation hearts (0.11±0.26, protocol I) (p=0.83). These findings indicated that endocardial sources might not have impact on the epicardial wavebreaks in this model. (p 14, para 3)

Minor:

6. Why the authors use such long (3.85ms) exposure time in optical mapping experiments? Will this enough to capture fast wave breaks? There are multiple papers reporting optical mapping in rabbit ventricle with 1k and even more sampling frequency with sound signal to noise ratio.

Response:

In previous studies, we have demonstrated that a similar camera system can successfully capture wavebreaks and phase singularities. The current study used an identical camera setting for simultaneous acquisition of wavebreaks from both sides of the myocardium. Furthermore, a long exposure time (3.85 s) is essential to capture the whole process from the beginning of burst pacing to the initiation of VF in optical mapping setting. We have addressed this concern in the method section as

“In previous studies, we have demonstrated that a similar camera with this frame rate system can successfully capture wavebreaks and phase singularities.13, 14 Furthermore, a long exposure time (3.85 s) is essential to capture the whole process from the beginning of burst pacing to the initiation of VF in optical mapping system.” (p 5, para 2 to p 6, para 1) 

7. Please clarify how 112 optical /p-ECG recordings are collected, at what time, right after pacing applied or wait for some time? Are they all the same regarding to the arrhythmogenesis? Some vulnerable time window may have larger arrhythmia possibility.

Response:

We have clarified this point by adding “During the burst pacing period in each VF inducibility test, both optical and p-ECG data were simultaneously recorded.” in the method section. (p 7, para 1)

We have added a table to show the PCLs and results (arrhythmogenesis) of each pacing attempt in the 10 hearts of the rotigaptide group. (supp Table 1)

8. Figure1A looks weird, the conduction velocity looks much smaller than the ones showed in Figure 1B.

Response:

Details of the CV evaluation have been described in the method section as “Briefly, epicardial activation perpendicular to the propagating wavefront was selected to measure the CV. The epicardial CV was evaluated by dividing the distance between 2 epicardial points with the conduction time (CT) using depolarization isochronal maps (Fig 1A). The CT between 2 epicardial points was measured by 50% crossover of the action potential amplitude in activation maps. We evaluated the CV at the centers of the anterior (A) and posterior (P) aspects of both ventricles (RV and LV). The mean of the CVs from these 4 areas became the CV of the heart.” (p 8, para 2)

In Fig 1A, we have added the CV values in each panel so that the readers will be easy to read and compare with Fig 1B. (Fig 1A) 

MS Number: PONE-D-19-26993R1

Responses to Reviewer #2

Comments to the Author

The submitted manuscript examined the susceptibility for pacing induced VF (PIVF) in isolated rabbit hearts under conditions of hypothermia (TH) and TH + Rotigaptide. TH led to a reduction in conduction velocity and prolongation of the APD, compared to baseline (37C.) The gap junction modifier Rotigaptide increased the conduction velocity, and reduced the susceptibility for PIVF. Similarly, Rotigaptide led to reduced ECG divergence and wavebreaks. There was a correlation between divergence and wavebreaks.

While the results are very interesting, and increased divergence and wavebreaks were associated with PIVF, it is difficult to assess the reliability of the results, as there may be other underlying confounders. Clarification of the conditions for the analyses and whether the results are from all animals (lumped together) or from specific animals that did vs did not develop PIVF, will help indicate whether divergence and wavebreaks may be markers for PIVF. While the introduction, discussion, goals of the study, and conclusions are about which patients will respond to anti-tachycardia pacing, the present results are not studying the success of anti-tachycardia pacing, but rather the susceptibility for PIVF. Therefore, the results do not support the conclusions.

Response:

We have added a table to show the PCLs and results of each pacing attempt in the 10 hearts of the rotigaptide group. (supp Table 1) With this table, the readers will be easy to tell how many pacing episodes came from each of the 10 animals.

We thank the reviewer for this insightful comment. Since ATP was only delivered to a pre-existing VT, the current experimental design was not compatible with ATP therapy. Therefore, we used programmed ventricular stimulation (PVS) instead of ATP to describe the burst ventricular pacing protocol, and to correlate the beat-to-beat morphological variation (divergence) with epicardial wavebreaks during TH. Accordingly, we have revised the introduction, discussion, and clinical implication sections to avoid over-statement. (p 3-4; p 15, para 2 to p 16, para 1; p 19, para1) 

ATP is performed to stop an arrhythmia, and not performed during sinus rhythm or quiescence. Thus, as this study is assessing the susceptibility and ECG dynamics (wavebreak & divergence) for PIVF, this study did not investigate or provide results to indicate which patients will respond successfully to ATP.

Response:

We thank the reviewer for this insightful comment. Since ATP was only delivered to a pre-existing VT, the current experimental design was not compatible with ATP therapy. Therefore, we used programmed ventricular stimulation (PVS) instead of ATP to describe the burst ventricular pacing protocol, and to correlate the beat-to-beat morphological variation (divergence) with epicardial wavebreaks during TH. Accordingly, we have revised the introduction, discussion, and clinical implication sections to avoid over-statement. (p 3-4; p 15, para 2 to p 16, para 1; p 19, para1) 

As there were inter-animal differences in the pacing frequency and number of pacing episodes, results need to be more standardized, without these underlying confounders. For example:

-Was PIVF achieved in all rabbits? If not, then do divergence and wavebreak serve as markers for animals that will develop PIVF or PIVF episodes about to arise? If not every rabbit developed PIVF, please present the number or percent of rabbits that developed PIVF (not episodes.)

Response:

We have added a table to show the PCLs and results of each pacing attempt in the 10 hearts of the rotigaptide group. (supp Table 1) With this table, the percentage and number of rabbits that developed PIVF and non-PIVF would be easy to read.

-When presenting results on wavebreak and divergence, clarify whether the results are from PIVF vs. no PIVF animals. For example, are no PIVF episodes only from animals that did not ever develop PIVF, or just an individual pacing episode that didn’t result in PIVF? If it is the later then the results in the study do not indicate which animals will develop PIVF (or success of ATP), but rather a pacing protocol that will not lead to PIVF.

Response:

We have added a table to show the PCLs and results of each pacing attempt in the 10 hearts of the rotigaptide group. (supp Table 1) A no PIVF episode indicated a pacing attempt that did not lead to VF. Only 2 hearts (#1 and #6) did not develop any PIVF episode throughout the pacing protocol. We have clarified this point in the result section as “Two hearts (#1 and #6) did not develop any PIVF episode throughout the pacing protocol.” (p 11, para 2)

-As each animal was exposed to 5-10 episodes of PIVF, results should be presented per rabbit, as some rabbit results (i.e., those with 10 pacing episodes) artificially skew the data compared to those with only 5 pacing episodes.

Response:

We have added a table to show the PCLs and results of each pacing attempt in the 10 hearts of the rotigaptide group. (supp Table 1) With this table, the readers will be easy to tell how many pacing episodes came from each of the 10 animals.

-Fig. 2, how many pacing episodes came from each of the 10 animals? Was there a relation between PIVF probability and pacing frequency?

Response:

We have added a table to show the PCLs and results of each pacing attempt in the 10 hearts of the rotigaptide group. (supp Table 1) With this table, the readers will be easy to tell how many pacing episodes came from each of the 10 animals. 

In this study, we observed that the PCL used in PIVF episodes (196±37 ms, n=51) is shorter than that in non-PIVF episodes (212±45 ms, n=61, p=0.044). We have included this data in the result section. (p 11, para 2)

-Fig. 3 shows snapshots of high vs. low divergence, but these are from different animals (rabbit#10 vs. rabbit#7) and possibly different pacing frequencies. Thus, it is unclear if the decrease in divergence was due to Rotigaptide, pacing frequency, animal, or another confounder.

Response:

We have revised this figure by choosing the same PCL (240 ms) in the same heart #10 to show the high/low divergence examples. (Fig 3) 

-Fig. 4 shows 112 pacing episodes (PIVF n=51, no PIVF n=61), but it is unclear how many came from each animal, and what were the pacing frequencies. Thus, it is not clear if differences in divergence and wavebreak is purely due to PIVF vs. no PIVF. Similarly, there is not a 1-to-1 relation of pacing episodes at TH vs. TH+Roti. Thus, it is unclear if Rotigaptide reduced divergence and wavebreak, or it was due to another confounder. Please provide percent change at TH+Roti compared to TH in the same animal and pacing frequency.

Response:

We have added a table to show the PCLs and results of each pacing attempt in the 10 hearts of the rotigaptide group. (supp Table 1) 

We also included the PCL data in the result section as “The PCLs used for PIVF induction were similar both before (206±40 ms, n=62) and after (204±46 ms, n=50) rotigaptide during TH (p=ns).” (p 11, para 2)

-Fig. 6 and 7 – If you want to indicate differences in waveform, divergence, and wavebreak in the presence (Fig. 6) and absence of Rotigaptide (Fig 7), then they need to be performed under the same conditions (i.e., same animal and pacing frequency), and not pacing at 220 ms vs. 280 ms.

Response:

We have revised these 2 figures by choosing the same PCL (200 ms) in heart #7 in the presence (new Fig 6) and absence (new Fig 7) of rotigaptide.

-Fig. 8 indicates the relationship between wavebreak and divergence, which is to be expected. It would be interesting to assess the correlation between wavebreak and divergence points from animals and episodes that only did vs. did not develop PIVF. Importantly, with proper analyses, these results could potentially indicate a marker for PIVF.

Response:

We also evaluated the correlation between divergence and wavebreaks in PIVF and non-PIVF episodes separately. We observed that the correlation between divergence and wavebreaks also exists in PIVF (p<0.001, R2=0.199) and non-PIVF (p<0.001, R2=0.261) episodes. (p 13, para 3)

 

MS Number: PONE-D-19-26993R1

Responses to Reviewer #3

Comments to the Author

Hereby the authors describe a study trying to shed light into the functional mechanisms of ATP therapy failure when high beat-to-beat morphological ‘dispersion/divergence’ on ventricular electrograms is present. Their hypothesis is that epicardial wavebreaks on the heart surface correlate to pacing induced morphological heterogeneity and predicts higher inducibility of ventricular arrhythmia. To test their hypothesis the investigators propose a study using optical mapping on the isolated Langendorff-perfused rabbit heart model, defining a protocol to test VF inducibility after hypothermia (30ºC, 30’) and, subsequent, Rotigaptide treatment (300nM, 20’). Despite describing their major limitations a critical concern is that the model and protocol presented by the authors may present bias not allowing them to support their conclusions and clinical implications; and, further, extrapolate to conditions with real ischemic or other substrate scenarios where ATP therapy success or failure is critical.

Response:

We thank the reviewer for this insightful comment. Since ATP was only delivered to a pre-existing VT, the current experimental design was not compatible with ATP therapy. Therefore, we used programmed ventricular stimulation (PVS) instead of ATP to describe the burst ventricular pacing protocol, and to correlate the beat-to-beat morphological variation (divergence) with epicardial wavebreaks during TH. Accordingly, we have revised the introduction, discussion, and clinical implication sections to avoid misunderstanding or over-statement. (p 3-4; p 15, para 2 to p 16, para 1; p 19, para1)

We have emphasized that the current experiment was performed in normal ventricles, and applying the study result to ischemic or other substrate should be with caution in the limitation section as below:

“Firstly, we used this hypothermic model to elucidate the correlation between ventricular divergence and epicardial wavebreak during PVS in normal ventricles. Whether our experimental conclusions can be extended to other experimental models (ischemic or heart failure models) remains to be verified.” (p 19, para 2)

Criticisms:

- The authors claim that divergence (morphological variation of electrical activity during pacing) correlates significantly to epicardial wavebreaks in the surface, however, it is clear from their adjusted R-squared goodness of fit, and figure 8, that the relationship between divergence and epicardial wavebreaks seems to poorly predict the data. Thus the authors should be cautious with their conclusions and discuss the implications carefully even if their claim is only that a significant relationship may exist, and being just one regressor, may have some average predictive power.

Response:

We agree with the reviewer that the relationship between divergence and epicardial wavebreaks seems to poorly predict the data because of the low R square value. Therefore, we have removed the term “statistically significant” and describe that “Linear regression analysis showed a correlation existed between ventricular divergence and the numbers of epicardial wavebreaks…..” in the result section. (p 13, para 3)

We also mentioned that “Although linear regression showed a correlation between ventricular divergence and the numbers of epicardial wavebreaks, the low R square value indicated that divergence along might not strongly predicted the wavebreaks numbers. Interpreting this data should be with caution.” in the limitation section. (p 20, para 1)

- Even though the authors have published the concept of ‘divergence of an electrical signal’ before, does not seem sound when speaking of morphological correlation or dispersion of signal patterns. Divergence is a concept used to represent the volume densities of the outward flux of vector fields when applied in the context of scalar fields such electrical or optical fields. Therefore, it may accurately be used to describe singularities in the heart surface, yet confusing when applied to the pseudo-ECG or ECG signals, or any other signal parameter dispersion. Instead, a classic correlation or entropic measure, deviation, or even turbulence from each beat could also represent significantly the instability of activation during susceptible induction.

Response:

We agreed with the reviewer that the concept of divergence is commonly used to describe the volume densities of the outward flux of vector fields. However, “divergence” is also used for measuring the "distance" of one probability distribution to the other in statistics, which is also known as “relative entropy”. We employed the latter concept to analyze the degree of morphologic variation of each activation on pseudo-ECG (p-ECG) in this study. Specifically, our divergence evaluated the degree of deviation of p-ECG and a reference activated waveform, where the averaged of 2-norm distance between each p-ECG and the template is calculated. Note that the template is regarded as the reference activated waveform and constructed by averaging all the activations on the p-ECG.

In order to make the concept of divergence clear, the following sentences were added to the revised manuscript with a reference cited (ref 20). "Note that the term of divergence is also known for the measure of the "distance" of one probability distribution to the other in statistics, which is also known as “relative entropy”. We employed the concept to analyze the degree of morphologic variation of each activation on p-ECG in this study." (p 9, para 3)

- Different studies (including the authors) have published before how arrhymogenic substrates may be favored by 30º-hypothermia, facilitating VF formation, alternans and wavebreaks. However, other authors, such as, Harada et all AJP Heart Circ Physiol 2008, or, Chorro FJ et al (AJP Heart Circ Physiol 2002), have provided strong evidence suggesting the antiarrhythmic therapeutic potential of moderate hypothermia, by also promoting self-termination of VT/VF. Please discuss. The authors only found a mild reduction in wavebreak formation under Rotigaptide treatment up to the onset of induction. Which was the lifetime of the wavebreaks (as measured) under each condition? Were they monitored only coupled with pacing beats? Was the duration or amount of wavebreaks directly related to the induction frequency or degree of morphological dispersion? Please discuss.

Response:

We have added a paragraph discussing different temperatures (moderate (33°C) and severe hypothermia (30°C)) on arrhythmogenesis from the studies of Harada et al, Chorro et al, and us as below:

“TH has been shown to improve neurological outcomes in patients resuscitated from cardiac arrest. Chorro et al reported that acute reduction of ventricular temperature to <20°C might simplify VF activation patterns and terminated VF. However, the temperature (<20°C) they used is far below the therapeutic range of hypothermia. Harada et al found that TH at 30°C in 2-dimensional rabbit ventricles enhanced wavebreaks and regeneration of new spiral waves during VF, facilitating ventricular tachycardia (VT) and VF maintenance, which is consistent with the pro-arrhythmic model used in the current study. In contrast, TH at 33°C might increase the chance of spiral wave collision and VT/VF self-termination, indicating that 33°C is a safe temperature for clinical use.” (p 16, para 2)

We have clarified that wavebreaks were monitored only coupled with pacing beats in the method section as “During the burst pacing period in each VF inducibility test, both optical (for wavebreak evaluation) and p-ECG (for divergence) data were simultaneously recorded.” (p 7, para 1)

- The Rotigaptide dose used in the study is quite high. What are the dose-response effects of Rotigaptide on induction mechanisms (divergence-wavebreaks)? Does it have antiarrythmic properties at higher concentrations? How is it modified in presence of anatomical substrate in presence of heterogenous tissue, for example, such as the presence of RF lesions, which may be more relevant to ATP therapy testing?

Response:

Previous animal studies suggested that the effective concentration of rotigaptide was 10-300 nM (ref 18). In this study, we chose the maximal effective concentration (300 nM) to evaluate the divergence-wavebreaks relationship in normal ventricles as in our previous study (ref 11). Whether the positive correlation observed in this study could be applied to rotigaptide at other concentration or in the presence of heterogeneous structures remains unclear. We have added this point in the limitation section as 

“Fourthly, we chose the maximal effective concentration (300 nM) to evaluate the divergence-wavebreaks relationship in normal ventricles. Whether the positive correlation observed in this study could be applied to rotigaptide at other concentration or in the presence of heterogeneous structures remains to be explored. (p 20, para 1)

- Including computer simulations may benefit the study testing different scenarios to prove the author’s hypotheses without intrinsic experimental limitations, and further, inexpensively. For example, the authors could test different pacing conditions including obstacles, regional differences in conductivities, distances to the obstacle, etc and controlling the experimental conditions. Have the authors determined before which cut-off ‘morphological divergence/dispersion’ values results in true ATP-failure or greater VT/VF inducibility probability?

Response:

We agree with the reviewer’s comment that computer simulation would have much benefit to control experimental conditions and prove our hypothesis inexpensively. However, our current software was not allowed to performed delicate computer simulation at this time. We have included this point in the limitation section as “Using computer simulation would also be beneficial in controlling experimental conditions and prove our hypothesis in the future.” (p 19, para 2)

 Our previous study (ref. 7) found that a lower divergence of the VT electrograms (cutoff value 0.73) predicted a successful ATP with a sensitivity and specificity of 81.9% and 65.9%, respectively. We have included this data in the introduction section as “By analyzing the electrograms from the anti-tachycardia pacing episodes in the ICD, we also found that high ventricular divergence is predictive of a failed therapy with a sensitivity and specificity of 81.9% and 65.9%, respectively.” (p 3, para 2)

- A major concern is the protocol. Is the heart submerged during the hypothermia protocol? Did you monitor the temperature during the protocol in the bath and heart surface? Why didn’t the authors include a control group (same time-stamped registers with no treatment) to rule out experimental constraints over time?

Response:

In this system, the hearts were superfused (submerged) in a thermostatized tissue bath. (p 5, para 1) 

We also revised the method section as “During the cooling procedure, the temperature in the upper, middle, and lower thirds of the tissue bath was checked every 1–2 min until 30°C was achieved at all levels. After reaching the target temperature (30°C), the tissue bath was kept at this temperature for an additional 5 min to ensure thermal homogeneity, before starting the study protocol.” (p 6, para 2) This cooling protocol was the same as those in our previous studies (ref 11-13).

We have added a new protocol (protocol II) using saline instead of rotigaptide in 4 hearts to serve as the control group. (p 7, para 2)

- While the idea behind the study may potentially be interesting, ‘to correlate beat-to-beat morphological dispersion during ATP failure to actual heart surface dynamics, and specifically to wavebreaks, under hypothermia’, the longitudinal protocol design, testing inducibility by burst pacing (only accounting to acute temperature-related modifications) make the results presented very limited. Severe hypothermia as shown by the authors before may be arrhythmogenic, which means that under fast pacing re-excitation and unidirectional block may appear leading under such substrate conditions to sustained or short-lived rotating activity. The question would be how early the divergence measure is sensitive and starts to accurately correlate to short-lived wavebreaks (when ventricular arrhythmia is not yet initiated). Otherwise, the relationship is well known already. Scenarios shown in figures 6 and 7 could shed some light in to it, however it is not clear in B weather optical and electrical signals temporarily match, and if so, how the extra beats 16-17-18 are related? They are not represented in C-D beat-to-beat single measures. Animated movies or videos illustrating the process under each condition would help interpretation and readability.

Response:

We matched the optical/p-ECG data by 2 ways: (1) we synchronized the clocks of these 2 recording systems so that the start time of each pacing train could be easily identified. (2) Since we started each pacing train during sinus rhythm, the onset morphological change on ECG and optical recording indicated the start of pacing train. (3) The p-ECG was continuously recording throughout the experiment, so the full-length p-ECG could be matched to the segmental optical data. 

We have provided the animated movies of Fig 7 to clarify the data interpretation for the readers. (supp movie)

- In figure 8: Please consider incorporating the association under each specific condition. Is the presumed significant correlation frequency of induction-dependent?

Response:

In the episodes of PCL≤200, the correlation between divergence and wavebreak was significant (p<0.001, R2=0.313). In the episodes of PCL>200, the correlation between divergence and wavebreak was also significant (P<0.001, R2=0.224). Therefore, this correlation is not frequency of induction dependent. (p 14, para 1)

- The authors use epicardial wavebreaks as a key measure of arrhythmia susceptibility or arrhythmia promoting and driving sources; yet wavebreaks measured as singularities by visual inspection, may represent sustained reentry after unidirectional block, simple to complex wavefront collisions from distant driving activity or even fibrillatory conduction. First, the methodology to track those points is not detailed enough, are detected every frame? In that case, the authors could provide detail information on the spatiotemporal properties comparing PIVF and No PIVF, both before and after Roti treatment. Which is the mechanism of initiation under each of the experimental conditions tested? Could it be possible that the mechanism is frequency of induction dependent, and therefore, better predicted or not by morphological variation on the pacing pseudo-ECG?

Response:

We tracked the phase singularities (PS) frame by frame manually. The life spans of each phase wavebreak during PIVF episodes were similar before (102±48 ms) and after (91±32 ms) rotigaptide treatment (p=0.50). (p 12, para 2) 

During burst ventricular pacing, we observed that unidirectional block and phase singularity occurred soon with sustained reentry degenerated into complex wavefront collisions and fibrillatory conduction. We have added this observation in the result section as: 

“Unidirectional block and wavebreak occurred soon during pacing with sustained reentry degenerated into complex wavefront collisions and fibrillatory conduction.”. (p 13, para 2) 

Supplementary move also showed this phenomenon. (supp movie)

Minor comments:

- How long lasted the whole protocol? Please provide a clarifying figure.

Response:

We have added a new figure to clarify the study protocol and the duration of each experiment. (supp Fig 1) 

The experiment duration of each heart was 126±15 min. (p 11, para 2)

- Through the study the authors use APD70 as a measure of APD. At least APD80 should be measured in optical mapping studies.

Response:

We have re-analyzed the APD using APD80 instead of APD70. (p 8, para 2) The APD restitution curve was also re-plotted using APD80. (Fig 1C, 1D) The revised data was shown in the result section. (p 11, para 1)

- How were the pseudo-ECG signals processed before applying the proposed ‘divergence’ equation?

Response:

The pseudo-ECG (p-ECG) was recorded and pre-processed according to our published article (ref 7). We have added the process method as below: 

“The maximal activation (local peak) of each p-ECG was identified firstly. Each p-ECG was then normalized with respect to the maximum values. The length of each p-ECG was defined by 80% of the average duration between the two consecutive activations. Each p-ECG was then extracted and aligned by the point of maximal activation so as to prevent the morphologic variation from the interference caused by the varying interval of two consecutive activations and the varying signal strength. Finally those aligned waveforms were applied to the proposed divergence equation.” (p 9, para 2)

- Methods section should be revised for inaccuracies and, potentially, partially moved to a supplementary section referencing previous studies when possible.

Response:

We have moved a table (supp Table 1), figure (supp Fig 1) and animated movie (supp movie) to the supplemental section.

- Figure2. Please complete information including SD/SE bards. Details on how VF episodes were considered should be included in the methods section, even include an episode in figure 2. Furthermore, which were the entry frequency and mean VF dominant frequency under each condition across the study sample.

Response:

To show the complete information of the 112 PIVF/non-PIVF episodes, we have included a new table to show the PCLs and results of each pacing attempt in the 10 hearts of the rotigaptide group. (supp Table 1) 

We also described how VF episodes were considered in the methods section as “Five to ten burst pacing trains (30 sec in duration) using the shortest PCL for ventricular capture were delivered to test the inducibility of VF. If VF could be induced and persisted for >1 min, a defibrillation shock would be delivered through the defibrillation coil, and this episode was considered to be a PIVF episode.” (p 7, para 1) 

We also added one non-PIVF (Fig 6) and PIVF (Fig 7) episodes as examples.

We also included the entry frequency (PCL) and mean VF dominant frequency data in the result section as “The PCLs used for PIVF induction were similar both before (206±40 ms, n=62) and after (204±46 ms, n=50) rotigaptide during TH (p=ns).” (p 11, para 2) and “In the PIVF episodes, the dominant frequency of the VFs were indifferent before (6.7±1.5 Hz, n=34) and after (6.2±1.1 Hz, n=17) rotigaptide infusion (p=0.20).” (p 11, para 2)

- Being only n=10 animals, including the individual data onto the bars would benefit the interpretation of the data.

Response:

We have added a table to show the PCLs and results of each pacing attempt in the 10 hearts of the rotigaptide group. (supp Table 1)

- In figure 3, different PCLs are used so the interpretation, why? How different were the baseline spontaneous frequencies at the beginning of the protocol and before rotigaptide treatment. How long was the protocol?

Response:

We have renewed the Fig 3 by using 2 episodes with the same PCL (240 ms), so that the reader will be easier to interpret. (Fig 3)

The cycle lengths of baseline spontaneous beats during TH were 1153±262 ms at the beginning of the protocol, and 1182±385 ms before rotigaptide treatment. (p10, para 3)

We have added a new figure to clarify the study protocol and the duration of each experiment. (supp Fig 1) The experiment duration of each heart was 126±15 min. (p 11, para 2)

- In figure 4, WB/phase map are compared when pacing induced VF, and, under Roti treatment. How were WB defined? Are they normalized? Why mean values are less than 1? Perhaps singularities should be measured automatically in each map, per temporal and physical unit in the whole field of view.

Response:

We have added the definition of WB and phase singularity (PS) in the method section as “A PS is defined as a site with an ambiguous phase surrounded by pixels exhibiting a continuous phase progression from –π to +π. Because of the close spatiotemporal correlation between PSs and wavebreak, PS has been a robust alternate representation of wavebreaks, serving as the source of VF.” (p 8, para 3)

Since the PS was evaluated from the beginning of burst pacing at sinus rhythm, the wavebreaks might take time to occur, leading to a mean value of less than 1. This is consistent with our previous studies. (ref. 12)

We agree with the reviewer’s comment. Although manual calculation of PSs is time-consuming, but can more accurately identify the PSs without numerical errors as in our previous studies. (ref. 17)

- Pacing cycle lengths used in the figures 6-7 are not reflected in the CV restitution curves why? Frequencies used for the burst pacing protocol are not clearly specified in the methods section.

Response:

We have added CV restitution data at PCL 220 ms (Fig 6) and 280 ms (Fig 7) in Fig 1B. 

We have added descriptions to clearly specify the frequencies used for the burst pacing protocol as “At baseline (37°C), we progressively shortened the S1 PCL (frequency) to identify the shortest PCL which achieved a 1:1 ratio ventricular capture. This PCL was selected as the initial PCL for burst pacing to induce VF.” in the method section. (p 6, para 3)

Also, we have included a table to show the PCLs and results of each pacing attempt in the 10 hearts of the study group. (supp Table 1)

- Differences in ‘divergence’ o measured ‘WB’ values seem rather small over time and the sample size limited for some comparisons. Which were your expected effect size and statistical power to estimate that sample size?

Response:

To achieve a medium effect size of 0.15 and a power of 0.8 in linear regression, the predictive sample size is 55. In this study, we used 112 pairs of divergence/wavebreak, which exceeded the predictive sample size. We have added this part in the statistics section. (p 10, para 2)

---

## [Editor Report · Decision Letter 1]

15 Jan 2020

PONE-D-19-26993R1

Ventricular Divergence Correlates with Epicardial Wavebreaks and Predicts Ventricular Arrhythmia in Isolated Rabbit Hearts during Therapeutic Hypothermia

PLOS ONE

Dear Dr. Hsieh,

Thank you for submitting your manuscript to PLOS ONE. After careful consideration, we feel that it has merit but does not fully meet PLOS ONE’s publication criteria as it currently stands. Therefore, we invite you to submit a revised version of the manuscript that addresses the points raised during the review process (see below).

We would appreciate receiving your revised manuscript by 01/21/2020. To enhance the reproducibility of your results, we recommend that if applicable you deposit your laboratory protocols in protocols.io, where a protocol can be assigned its own identifier (DOI) such that it can be cited independently in the future. For instructions see: http://journals.plos.org/plosone/s/submission-guidelines#loc-laboratory-protocols

We look forward to receiving your revised manuscript.

Kind regards,

Manuel Zarzoso, Ph.D.

Academic Editor

PLOS ONE

Additional Editor Comments (if provided):

The authors, overall, have addressed most of the concerns raised by the reviewers. The reach of the conclusions based on the experimental model used, has been toned down. Furthermore, new experimental series (II, III and IV), additional results and modifications of the already presented, were added, which have contributed to improve the paper from its previous version, thus addressing methodological questions that could limit the validity of the results. The number of experiments in protocol III could have been higher, since n=2 hearts seems a bit low in order to perform statistical analyses (even with non-parametric tests, and even using several recordings per heart). On the other hand, those aspects that could not been addressed with experimental data, were included as limitations.

I have, however, some minor concerns:

1. I would suggest removing the sentence in Methods (page 10, paragraph 2 "To achieve a medium effect size of 0.15 and a power of 0.8 in linear regression, the predictive sample size is 55".), since it is already clear that the model does not provide high predictive power with the determination coefficient (R2), as it is also stated in the text.

2. The authors should review the text and correct text editing (i.e. page 7 last line, should read “H₂O” with 2 in subscript instead of “H2O”; p13, par2 should read “...and fibrillatory conduction (movie S1).” instead of “…and fibrillatory conduction. (Supp movie 1)”; p14 para1 “R²” with 2 in superscript instead of “R2”, among many others).

3. Referencing should be revised, the number in brackets should be written before the period “[1].” And not after “.[1]”

4. Modify the sentence in p17 par1 to “These findings suggest that ventricular divergence on ECG could be a predictor of epicardial wavebreaks during PVS”. P17 par2 revise the meaning of “In multiple wavelet hypotheses, wavebreaks resulting from collisions of propagation wavelets and generations of new wavelets are the basis for VF maintenance.[27] Using optical mapping techniques, PS found on phase maps is the area of ambiguous activation state which underlies the formation of rotor and wavebreaks which serve as a source of VF.[10]”.

5. Soften the conclusion regarding the predictive value of ventricular divergence, i.e. “Ventricular beat-to-beat morphological variation evaluated as divergence was significantly correlated with, and could predict epicardial wavebreak during ventricular pacing at TH” in the abstract and conclusion sections.

6. Remove “*Fisher’s exact test” from Fig 2 and include the information in Statistical Analysis.

---

## [Author Response · Author response to Decision Letter 1]

20 Jan 2020

MS Number: PONE-D-19-26993R2

Responses to Editor

Comments to the Author

The authors, overall, have addressed most of the concerns raised by the reviewers. The reach of the conclusions based on the experimental model used, has been toned down. Furthermore, new experimental series (II, III and IV), additional results and modifications of the already presented, were added, which have contributed to improve the paper from its previous version, thus addressing methodological questions that could limit the validity of the results. The number of experiments in protocol III could have been higher, since n=2 hearts seems a bit low in order to perform statistical analyses (even with non-parametric tests, and even using several recordings per heart). On the other hand, those aspects that could not been addressed with experimental data, were included as limitations.

Response:

We thank the editor for these insightful comments. In protocol III, we have performed an experiment to increase the heart number (n=3) and improve statistical analyses. (p 7, para 3)

The addition of this new data did not change the main findings in protocol III as below: 

“Different Pacing Sites on the Divergence and VF Inducibility

The divergence of the PIVF episodes was not different between RV (0.15±0.04) and LV (0.15±0.07) pacing (p=0.97). Also, the divergence of the non-PIVF episodes was not different between RV (0.12±0.07) and LV (0.11±0.05) pacing (p=0.60). The inducibility of VF was also similar before (p=0.20) and after (p=0.23) rotigaptide infusion between RV and LV pacing. These findings indicated that the divergences and VF inducibility in this model were not affected by changing the pacing sites.” (p 14, para 1)

I have, however, some minor concerns:

1. I would suggest removing the sentence in Methods (page 10, paragraph 2 "To achieve a medium effect size of 0.15 and a power of 0.8 in linear regression, the predictive sample size is 55".), since it is already clear that the model does not provide high predictive power with the determination coefficient (R2), as it is also stated in the text.

Response:

We have removed this sentence in the method section. (p 10, para 2)

2. The authors should review the text and correct text editing (i.e. page 7 last line, should read “H₂O” with 2 in subscript instead of “H2O”; p13, par2 should read “...and fibrillatory conduction (movie S1).” instead of “…and fibrillatory conduction. (Supp movie 1)”; p14 para1 “R²” with 2 in superscript instead of “R2”, among many others).

Response:

We have corrected the text editing problems, including those mentioned by the editor, throughout the manuscript. (highlighted parts)

3. Referencing should be revised, the number in brackets should be written before the period “[1].” And not after “.[1]”

Response:

We have moved the “number in brackets” to be before the period throughout the manuscript.

4. Modify the sentence in p17 par1 to “These findings suggest that ventricular divergence on ECG could be a predictor of epicardial wavebreaks during PVS”. P17 par2 revise the meaning of “In multiple wavelet hypotheses, wavebreaks resulting from collisions of propagation wavelets and generations of new wavelets are the basis for VF maintenance.[27] Using optical mapping techniques, PS found on phase maps is the area of ambiguous activation state which underlies the formation of rotor and wavebreaks which serve as a source of VF.[10]”.

Response:

We have modified this sentence as the editor’s suggestion as “These findings suggest that ventricular divergence on ECG could be a predictor of epicardial wavebreaks during PVS.” (p 17, para 1)

The meaning of “multiple wavelet hypothesis” has also been revised as “In multiple wavelet hypothesis, wavebreaks resulting from collisions of propagation mother wavelets might generate daughter wavelets in a self-sustaining turbulent process to maintain VF [27]. Using optical mapping techniques, PS found on phase maps is the area of ambiguous activation state which underlies the formation of rotor and wavebreaks, and serves as a source of VF [10].” (p 17, para 2)

5. Soften the conclusion regarding the predictive value of ventricular divergence, i.e. “Ventricular beat-to-beat morphological variation evaluated as divergence was significantly correlated with, and could predict epicardial wavebreak during ventricular pacing at TH” in the abstract and conclusion sections.

Response:

We have revised the conclusions in the abstract as “Ventricular divergence correlated with, and might be predictive of epicardial wavebreaks during PVS at TH.” (p 2, abstract)

Also, we softened the main conclusion in the manuscript as “Ventricular beat-to-beat morphological variation evaluated as divergence was correlated with, and might be predictive of, epicardial wavebreak during ventricular pacing at TH.” (p 20, para 2)

6. Remove “*Fisher’s exact test” from Fig 2 and include the information in Statistical Analysis.

Response:

We have removed the “*Fisher’s exact test” from Fig 2, and included the information in Statistical Analysis as “Fisher’s exact test was used to compare the categorical data between groups.” (p 10, para 2)

---

## [Editor Report · Decision Letter 2]

24 Jan 2020

Ventricular Divergence Correlates with Epicardial Wavebreaks and Predicts Ventricular Arrhythmia in Isolated Rabbit Hearts during Therapeutic Hypothermia

PONE-D-19-26993R2

Dear Dr. Hsieh,

We are pleased to inform you that your manuscript has been judged scientifically suitable for publication and will be formally accepted for publication once it complies with all outstanding technical requirements.

With kind regards,

Manuel Zarzoso, Ph.D.

Academic Editor

PLOS ONE

---

## [Editor Report · Acceptance letter]

13 Feb 2020

PONE-D-19-26993R2 

Ventricular Divergence Correlates with Epicardial Wavebreaks and Predicts Ventricular Arrhythmia in Isolated Rabbit Hearts during Therapeutic Hypothermia 

Dear Dr. HSIEH:

I am pleased to inform you that your manuscript has been deemed suitable for publication in PLOS ONE. Congratulations! Your manuscript is now with our production department. 

With kind regards,

on behalf of

Dr. Manuel Zarzoso 

Academic Editor

PLOS ONE